# Learning to Focus: Causal Attention Distillation via Gradient-Guided Token Pruning

**Yiju Guo♪, Wenkai Yang♪, Zexu Sun♩ , Ning Ding♩, Zhiyuan Liu♩, Yankai Lin♪✉**

♪ Gaoling School of Artificial Intelligence, Renmin University of China

♩ Department of Computer Science and Technology, Tsinghua University

♩ Baidu Inc.

✉{yijuguo, yankailin}@ruc.edu.cn

## Abstract

Large language models (LLMs) have demonstrated significant improvements in contextual understanding. However, their ability to attend to truly critical information during long-context reasoning and generation still falls behind the pace. Specifically, our preliminary experiments reveal that certain distracting patterns can misdirect the model's attention during inference, and removing these patterns substantially improves reasoning accuracy and generation quality. We attribute this phenomenon to spurious correlations in the training data, which obstruct the model's capacity to infer authentic causal instruction–response relationships. This phenomenon may induce redundant reasoning processes, potentially resulting in significant inference overhead and, more critically, the generation of erroneous or suboptimal responses. To mitigate this, we introduce a two-stage framework called **Lea**rning to **F**ocus **(LeaF)** leveraging intervention-based inference to disentangle confounding factors. In the first stage, LeaF employs gradient-based comparisons with an advanced teacher to automatically identify confounding tokens based on causal relationships in the training corpus. Then, in the second stage, it prunes these tokens during distillation to enact intervention, aligning the student's attention with the teacher's focus distribution on truly critical context tokens. Experimental results demonstrate that LeaF not only achieves an absolute improvement in various mathematical reasoning, code generation and multi-hop question answering benchmarks but also effectively suppresses attention to confounding tokens during inference, yielding a more interpretable and reliable reasoning model. [2]

## 1 Introduction

Large language models (LLMs) have achieved remarkable success in natural language processing tasks, demonstrating strong capabilities in contextual understanding [57, 37, 4] and language generation [14, 11]. Despite these advancements, LLMs still struggle to maintain focus on truly critical information, especially in long-context reasoning [52, 55, 20] and complex instruction-based tasks [51, 31], which adversely impacts reasoning accuracy and generation quality.

To systematically investigate this phenomenon, we first identify distracting patterns via gradient-based comparisons of teacher and student sensitivities, then assess the performance of student models on NuminaMath-CoT [26] and AceCode-87K [53]. Surprisingly, as shown in Figure 1, simply pruning distracting patterns yields substantial gains—improving average accuracy by over 20% on the MATH

---

✉ Corresponding author: Yankai Lin (yankailin@ruc.edu.cn).

[2]Code and data are available at `https://github.com/RUCBM/LeaF`.

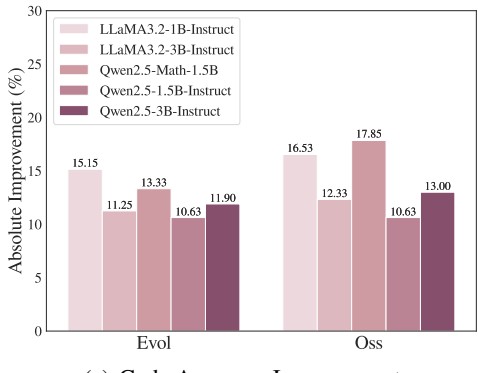

(a) Code Accuracy Improvement

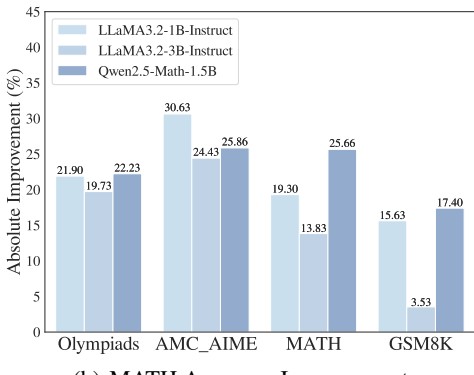

(b) MATH Accuracy Improvement

Figure 1: Accuracy improvements achieved by removing confounding tokens from small models on the math and code training corpora. The results demonstrate a significant increase in performance, with over 20% improvement on the math corpus and more than 10% on the code corpus. (For further details on these categories, see Appendix A.)

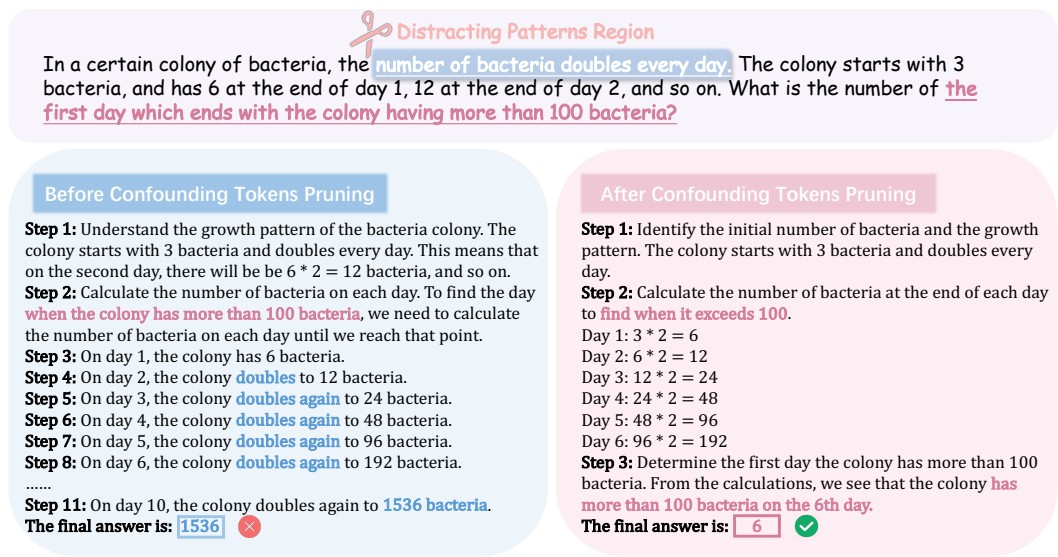

Figure 2: Comparison of reasoning before and after pruning distracting patterns. Blue-shaded regions indicate pruned confounding tokens. Pink highlights mark areas that require focus, while blue highlights show where excessive attention caused errors.

training corpus [26] and more than 10% on Code training corpus [53]. Furthermore, we observe greater improvements on AMC_AIME [26] compared to GSM8K [8], suggesting that complex reasoning problems may contain more distracting patterns that interfere with model inference. These findings demonstrate that mitigating the influence of distracting patterns is essential for improving the robustness and accuracy of LLM reasoning. We further present a representative case in Figure 2. Removing distracting patterns from the instruction, without any additional training, helps the model focus on critical information and enhance reasoning. This suggests that improving attention to relevant details is a promising direction for advancing model reasoning.

Based on our analysis, we adopt a causal perspective (Figure 3) to explain and mitigate the observed phenomenon. We propose **Learning to Focus** (**LeaF**), a two-stage framework that treats distracting patterns as spurious confounders in LLM reasoning. In the first stage, LeaF identifies confounding tokens through gradient-based comparisons between a high-capacity teacher and a student model. Then, it generates counterfactual samples by span pruning, removing contiguous spans of the detected confounding tokens from each instruction. In the second stage, LeaF introduces a hybrid distillation loss that minimizes two KL divergences: one for original sample (standard distillation) and one for

counterfactual sample (counterfactual distillation). This composite objective encourages the student model to capture true causal dependencies by contrasting the teacher model's outputs before and after pruning confounding tokens, improving the robustness and interpretability of LLM reasoning.

We demonstrate through comprehensive experiments that Learning to Focus (LeaF) significantly enhances critical token identification and attention consistency during inference, leading to improved performance on downstream tasks such as mathematical reasoning, code generation and multi-hop question answering. Compared to standard knowledge distillation, LeaF yields an average accuracy gain of 2.41% on GSM8K [8], MATH-500 [17], and OlympiadBench [16] for LLaMA-1B/3B-Instruct [34] and Qwen2.5-Math-1.5B [48]. In code generation, LeaF achieves an average improvement of 2.48% on HumanEval+ [32], LeetCode [9] and LivecodeBench [22]. On multi-hop question answering, LeaF improves performance by an average of 3.24% on HotpotQA [50], 2WikiMultiHopQA [19], and Musique [44]. These results validate our hypothesis that enhancing attention to key information is critical for improving reasoning performance. Attention visualizations in Section 4.4 further demonstrate LeaF's interpretability.

## 2 Methodology

### 2.1 Causal Framework

Following Pearl's Structural Causal Model [39], we formulate a DAG $G$ to model the causal relationships among different components (refer to Figure 3). We assume the distribution of $(X, A, Y)$ is faithful to the DAG. In this framework, $X = [x_1, x_2, \ldots, x_n]$ represents the input tokens, and $Y$ is the model's output. We define confounding tokens as a subset $A$ that obscure true causal relationships by introducing spurious correlations with both the output $Y$ and the complementary input $X \setminus A$. These misleading dependencies distort the model's attention mechanisms and bias its reasoning process, ultimately yielding unreliable predictions.

The desired behavior of the model is to **eliminate spurious correlations** introduced by confounding tokens $A$. When $A$ influences both $X$ and $Y$ (see dashed arrows in Figure 3), the observed conditional distribution becomes:

$$P(Y \mid X_i = x) = \sum_A P(Y \mid X_i = x, A) \, P(A \mid X_i = x), \quad (1)$$

which deviates from the interventional distribution $P(Y \mid \mathrm{do}(X_i))$ and reflects bias introduced by the indirect influence of $A$ on $Y$ through spurious paths.

To block these non-causal influences, we propose **causal pruning**, which removes the effect of $A$ prior to distillation. This encourages the student model to learn attention patterns grounded in true causal structure, improving robustness and interpretability.

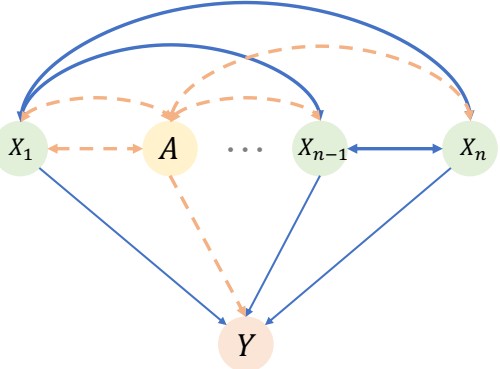

Figure 3: Causal graph of the reasoning process. $X$ represents the input prompt, and $Y$ denotes the model's output. A subset of tokens in $X$, identified as confounding tokens ($A$), introduces spurious correlations that disrupt the reasoning process. Our method detects and masks $A$, effectively eliminating the spurious edge from $A$ to $Y$ and restoring the true causal dependency.

### 2.2 LeaF: Learning to Focus Framework

To eliminate spurious dependencies, we introduce the **Learning to Focus (LeaF)** framework (Figure 4), which consists of two main stages: (1) Confounding Token Detection (Section 2.2.1), where LeaF identifies confounding tokens via teacher–student gradient-based comparisons and constructs counterfactual samples by pruning these tokens. (2) Causal Attention Distillation (Section 2.2.2), where LeaF captures causal dependencies through a hybrid distillation loss that aligns the student with the teacher on both original and counterfactual samples. We discuss them in detail in the following.

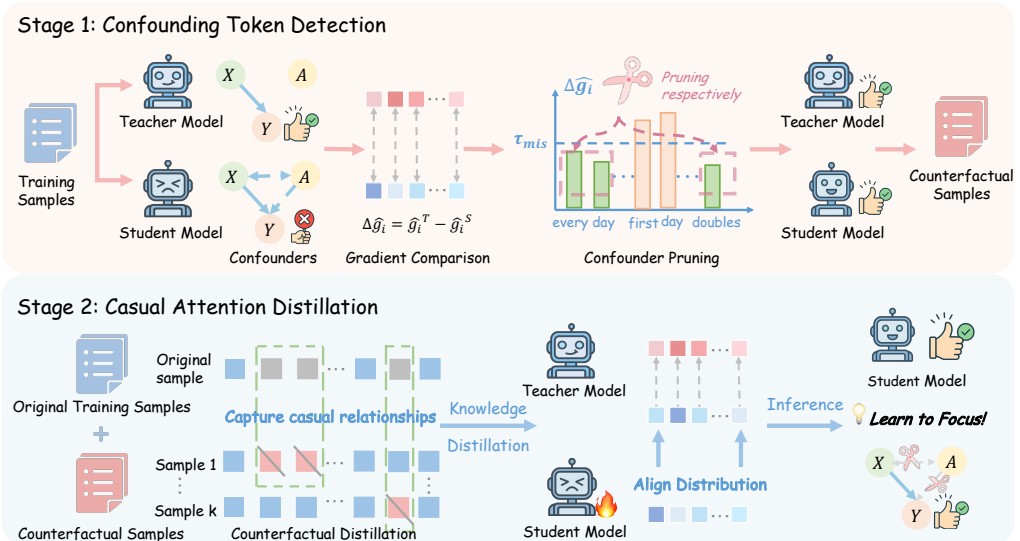

Figure 4: **Method Overview.** The training pipeline comprises two key stages: (1) **Confounding Token Detection**: gradient-based comparisons between an advanced teacher model and the student model are used to identify confounding tokens in the training samples and constructs counterfactual samples by pruning these tokens; and (2) **Causal Attention Distillation**: prune identified confounders respectively during training to align the student's attention with the teacher's and capture casual relationships. This targeted intervention steers the model toward actual causal dependencies, enhancing both robustness and interpretability.

### 2.2.1 Confounding Token Detection

To identify the confounding tokens $A$ that introduce spurious correlations, we adopt a **gradient-based approach** [42, 43] to quantitatively measure the influence of each token on the model's output $Y$. Specifically, we leverage the gradient sensitivity of both the teacher model $\boldsymbol{\theta}_T$ and the student model $\boldsymbol{\theta}_S$. We focus on data instances mispredicted by the student but correctly handled by the teacher to isolate confounding tokens. For each token $x_i \in X$, we compute the gradient of the cross-entropy loss between predicted logits and gold references with respect to the embedding of $x_i$ for both models.

$$g_i^{(T)} = \left| \frac{\partial \ell(x_i \mid X; \boldsymbol{\theta}_T)}{\partial x_i} \right|, \quad g_i^{(S)} = \left| \frac{\partial \ell(x_i \mid X; \boldsymbol{\theta}_S)}{\partial x_i} \right|. \tag{2}$$

These gradients reflect the sensitivity of each model to perturbations in $x_i$. To enable token-level comparison between models with differing gradient scales, we apply min-max normalization to the sensitivity values.

$$\hat{g}_i^{(T)} = \frac{g_i^{(T)} - \min_j g_j^{(T)}}{\max_j g_j^{(T)} - \min_j g_j^{(T)}}, \quad \hat{g}_i^{(S)} = \frac{g_i^{(S)} - \min_j g_j^{(S)}}{\max_j g_j^{(S)} - \min_j g_j^{(S)}}. \tag{3}$$

To identify confounding tokens, we capture the difference in token-level attention between the teacher and student models by computing the gradient difference for each token:

$$\Delta \hat{g}_i = \hat{g}_i^{(T)} - \hat{g}_i^{(S)}. \tag{4}$$

To ensure consistent scaling of gradient discrepancies across instances, we normalize the gradient difference and classify a token $x_i$ as a **Confounding Token** if (i) it receives significant attention from the student model but negligible attention from the teacher during inference, as formalized in Equation 5, and (ii) its removal results in correct predictions from both models.

$$\frac{\Delta \hat{g}_i - \min_j \Delta \hat{g}_j}{\max_j \Delta \hat{g}_j - \min_j \Delta \hat{g}_j} \leq \tau_{\text{confounder}}, \tag{5}$$

where $\tau_{\text{confounder}}$ is a threshold determined via statistical analysis on a validation set. Sensitivity analyses on the threshold are presented in Section 4.3. Intuitively, confounding tokens capture sensitivity discrepancies between the teacher and student models that indicate spurious dependencies.

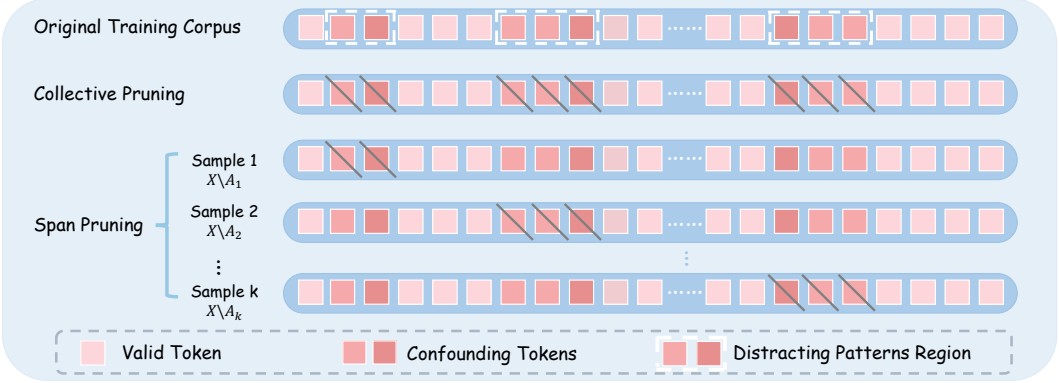

Figure 5: Illustration of Collective Pruning and Span Pruning.

Moreover, we also explored perplexity-based detection, but without teacher guidance, it tends to capture tokens indicative of model uncertainty rather than true confounders, especially on challenging tasks like the Olympiads. Accordingly, we adopt gradient-based detection as our primary strategy (see Section 4.1 for details).

**Pruning Strategies.** We study two pruning strategies for removing confounding tokens from instruction $X$: (1) **Collective Pruning**, which removes the entire set of identified confounders $A$, yielding $X \backslash A$. (2) **Span Pruning**, which removes only one contiguous confounding span $A_i$ at a time, yielding $X \setminus A_i$. Preliminary experiments show that span pruning outperforms collective pruning (see Appendix C), as pruning all distracting patterns simultaneously disrupts sentence integrity. Hence, we construct the counterfactual samples via the span pruning strategy:

$$\mathcal{D}_{\text{pruned}} = \left\{ (X \setminus A_i, y) \right\}_{i=1}^{k},$$

where each $A_i$ denotes a distinct confounding span. This augmentation encourages the model to learn reasoning paths invariant to specific confounders.

### 2.2.2 Causal Attention Distillation

After generating both original and counterfactual samples, we optimize two complementary distillation objectives to steer the student toward true causal dependencies:

**Standard Distillation.** Align the student's output distribution with the teacher's on original instructions:

$$\mathcal{L}_{kd} = D_{\text{KL}}\big( p_T(y \mid X) \,\|\, p_S(y \mid X) \big),$$

where $p_T$ and $p_S$ denote the teacher and student output distributions, respectively.

**Counterfactual Distillation.** Align the student's output distribution with the teacher's on counterfactual instructions $X \setminus A$ (confounders pruned):

$$\mathcal{L}_{cd} = D_{\text{KL}}\big( p_T(y \mid X \setminus A) \,\|\, p_S(y \mid X \setminus A) \big).$$

we blend these objectives with a weighting factor $\lambda$:

$$\mathcal{L} = \lambda\,\mathcal{L}_{kd} \,+\, (1 - \lambda)\,\mathcal{L}_{cd},$$

where $\lambda \in [0, 1]$ controls the trade-off between standard and counterfactual distillation. This composite loss steers the student to preserve semantic knowledge while enforcing genuine causal dependencies.

**Response Splitting Strategies.** For our methods, we consider two variants: (1) **Instruct-level Pruning**: We detect and prune confounding tokens only in the instructions, and perform our LeaF on the instruction-level pruned samples. (2) **Both Instruct- and Response-level Pruning**: We can also treat previously generated tokens as contextual input, and prune those that are misleading for subsequent generation, in order to help the model produce more accurate continuations. Thus, we detect and prune confounding tokens in both instructions and preceding generations.

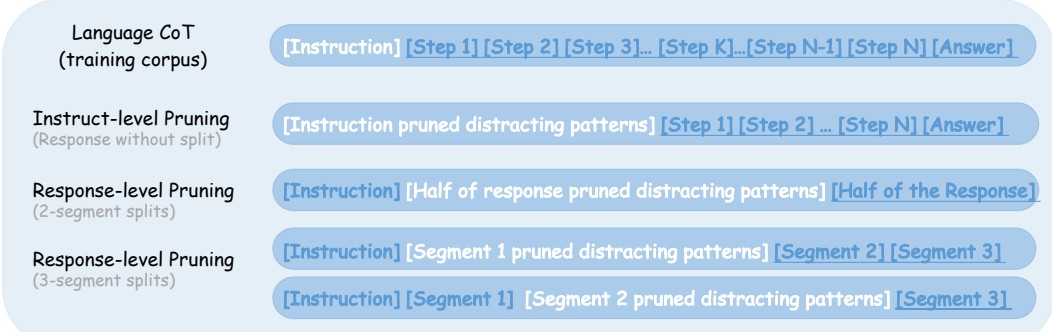

Figure 6: Illustration of Response Splitting Strategies: Language CoT, Instruct-level Pruning, and Response-level Pruning (2-segment and 3-segment splits). Highlighted white areas represent the input, and blue underlined areas represent the outputs used for cross-entropy loss computation.

## 3 Experiments

### 3.1 Experiment Settings

**Training Datasets.** We conduct experiments to evaluate the effectiveness of our proposed method Learning to Focus Framework (LeaF) on mathematical reasoning, code generation, and multi-hop question answering tasks. For mathematical reasoning, to ensure the model encounters an equal number of confounding tokens across tasks, we randomly select 30k instances from each of the following subsets in NuminaMath-CoT [26]: Olympiads [16], AMC_AIME [26], GSM8K [8], and MATH [17]. For code generation, we randomly select a subset of 120k instances from the AceCode-87k [53] dataset. For multi-hop question answering, we construct the training set by merging the KILT [40] datasets provided in Helmet [52], totaling 3k annotated samples drawn equally from HotpotQA [50], NQ [1], and PopQA [36], where each query is explicitly linked to its corresponding gold passages containing the answers.

**Evaluation Datasets.** For math task evaluation, we select three widely used benchmarks with varying levels of difficulties, including GSM8K [8], MATH-500 [17], and OlympiadBench [16]. In code domain, we conduct evaluations on HumanEval+[32], LeetCode[9], and LivecodeBench (v4)[22], providing a comprehensive assessment across diverse aspects of coding performance. For multi-hop question answering, we evaluate on three representative benchmarks, including HotpotQA [50], 2WikiMultiHopQA [19], and Musique [44]. The detailed description of the mathematical reasoning, code generation and multi-hop QA benchmarks is provided in Appendix K.

**Base Models and Baselines.** We conduct comprehensive experiments on two different model families, LLaMA family [38, 34] and Qwen family [48], covering various sizes of models. For LLaMA-based experiments, we employ LLaMA3.2-1B-Instruct [34] and LLaMA3.2-3B-Instruct [34] as student models, while their teacher models are set as LLaMA3.3-70B-Instruct [34]. For Qwen-based experiments, we use Qwen2.5-Math-1.5 [48] as the student model, with Qwen2.5-72B-Instruct [48] as the teacher model. Our baseline for comparison is standard knowledge distillation without pruning. We use the CoT-based variant for math and the vanilla variant for code generation and multi-hop QA.

**Training and Evaluation Settings.** (1) During training, models are trained using the Alpaca-LoRA framework with full-parameter logits knowledge distillation and a cosine learning rate schedule with a maximum learning rate of $10^{-5}$ for three epochs. The batch size is 64 for LLaMA-based models and 32 for Qwen-based models. Detailed hyperparameters and platform information are in Appendix J. (2) In evaluation, teacher and student models are evaluated on math, code and QA tasks using greedy decoding. The maximum generation length is 1024 tokens for code and 16,384 tokens for math tasks. Official chat templates are followed during inference. Full evaluation details are in Appendix B.

### 3.2 Main Results

The main results are presented in Table 1 and Table 2. We can draw several conclusions from the results: (1) For open-source baselines, both standard distillation and our approach lead to

Table 1: Performance on MathBench and CodeBench for Instruct models (LLaMA 3.2–1B/3B-Instruct) and Base model (Qwen 2.5–Math-1.5B) under three pruning schemes (no mask, Instruct-level Mask, and Response-level Mask), where Instr Mask refers to Instruct-level pruning, and Resp Mask refers to Response-level pruning. The best and second-best results are marked in **bold** and underlined, respectively.

| Model | MathBench | | | | CodeBench | | | |
|---|---|---|---|---|---|---|---|---|
| | GSM8K | MATH-500 | Olympiad-Bench | Avg. | Human-Eval+ | Leet-Code | Livecode-Bench | Avg. |
| ***Teacher Model*** | | | | | | | | |
| LLaMA3.3-70B-Instruct | 95.60 | 70.40 | 36.50 | 67.50 | 78.05 | 53.90 | 45.02 | 58.99 |
| Qwen2.5-72B-Instruct | 95.45 | 73.80 | 41.25 | 70.17 | 81.71 | 69.40 | 54.42 | 68.51 |
| ***LLaMA3.2-1B-Instruct*** | | | | | | | | |
| Instruct Model (Pre-KD) | 44.88 | 24.20 | 5.79 | 24.96 | 29.27 | **7.22** | 9.68 | 15.39 |
| KD w/o Mask | 56.79 | 33.40 | 8.90 | 33.03 | 32.32 | 6.11 | **13.74** | 17.39 |
| LeaF (Instr Mask) | 57.70 | **35.40** | **10.09** | 34.40 | 39.02 | 6.67 | 13.60 | 19.76 |
| LeaF (Instr & Resp Mask) | **58.98** | 35.20 | 9.94 | **34.71** | **39.63** | **7.22** | 12.48 | **19.77** |
| ***LLaMA3.2-3B-Instruct*** | | | | | | | | |
| Instruct Model (Pre-KD) | 76.88 | 42.80 | 13.20 | 44.29 | 48.78 | 13.89 | 20.34 | 27.67 |
| KD w/o Mask | 82.87 | 49.00 | 18.99 | 50.29 | 54.88 | 16.67 | 24.12 | 31.89 |
| LeaF (Instr Mask) | 83.09 | 51.80 | 20.77 | 51.88 | 55.49 | 19.44 | 25.39 | 33.44 |
| LeaF (Instr & Resp Mask) | **84.69** | **52.40** | **22.55** | **53.21** | 56.10 | **21.67** | **25.81** | **34.53** |
| ***Qwen2.5-Math-1.5B*** | | | | | | | | |
| Base Model (Pre-KD) | 65.20 | 41.40 | 21.96 | 42.85 | 35.37 | 6.67 | 1.26 | 14.43 |
| KD w/o Mask | 82.18 | 67.80 | 31.16 | 60.38 | 41.46 | 7.78 | 10.10 | 19.78 |
| LeaF (Instr Mask) | 84.69 | 68.60 | **32.79** | 62.03 | 42.68 | **9.94** | 10.80 | 20.97 |
| LeaF (Instr & Resp Mask) | **85.29** | **70.60** | 31.75 | **62.54** | 43.29 | **9.94** | **13.04** | **21.92** |

Table 2: Performance on multi-hop QA benchmarks under different pruning strategies. Given the short response length of multi-hop QA tasks, only the LeaF (Instruct-level Mask) variant is evaluated. Given the long-context requirement of multi-hop QA, we validate LeaF using LLaMA-3.2-3B-Instruct.

| Model | 2WikiMultiHopQA | | | Musique | | | HotpotQA | | | Avg. |
|---|---|---|---|---|---|---|---|---|---|---|
| | EM | F1 | SubEM | EM | F1 | SubEM | EM | F1 | SubEM | |
| ***LLaMA3.2-3B-Instruct*** | | | | | | | | | | |
| Instruct Model (Pre-KD) | 22.50 | 34.73 | 45.50 | 9.50 | 16.56 | 18.50 | 19.50 | 32.67 | 36.00 | 26.16 |
| KD w/o Mask | 43.50 | 51.93 | 53.00 | 20.50 | 28.56 | 22.50 | 39.00 | 49.30 | 43.00 | 39.03 |
| LeaF (Instr Mask) | **46.50** | **53.89** | **55.00** | **27.00** | **33.00** | **28.00** | **40.50** | **52.06** | **44.50** | **42.27** |

improvements in the performance of models across nearly all tasks in math, code and multi-hop question answering domains. (2) In terms of performance, LeaF consistently outperforms standard knowledge distillation when using the same training corpus. This suggests that LeaF has the potential to enable the model to attend to critical information more effectively, thereby enhancing its reasoning capabilities. (3) Extending from instruction-level to response-level pruning yields further performance improvements across most tasks in the LLaMA and Qwen series, suggesting that distracting patterns at the response level affect subsequent generations. We hypothesize that instruction-level and response-level distracting patterns differ, and simultaneously learning both can enhance the model's reasoning capabilities. A more detailed analysis based on the response splitting strategy is provided in Section 4.2. Additionally, Appendix E shows that LeaF is more robust than other pruning strategies.

## 4 Further Analysis

In this section, we conduct masking strategies analysis (Section 4.1), response splitting analysis (Section 4.2) and threshold sensitivity analysis (Section 4.3). Finally, we present a case study (Section 4.4) to illustrate the interpretability of our approach. In the appendix, we include robustness

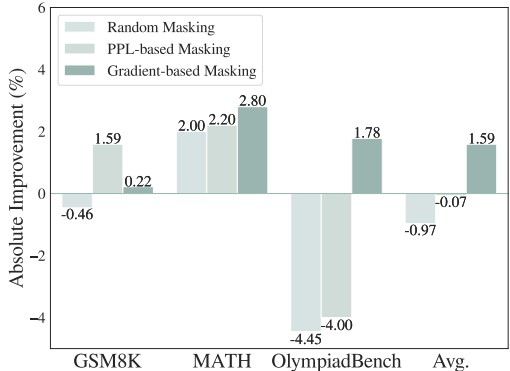
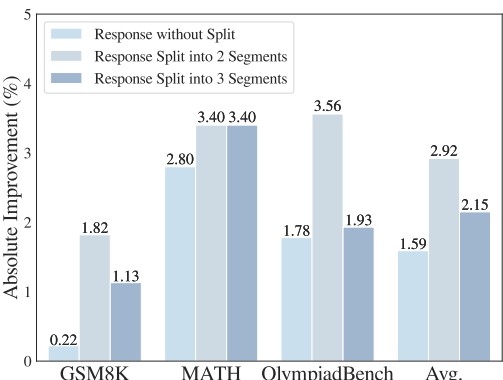

Figure 7: Comparison of accuracy improvement with masking strategies over baseline (KD).

Figure 8: Comparison of accuracy improvement with splitting strategies over baseline (KD).

analysis (Appendix E), threshold stability across tasks (Appendix G), computational overhead analysis (Appendix D), ablation study (Appendix F), and dataset statistics of counterfactual and original samples (Appendix H).

## 4.1 Masking Strategies Analysis

To demonstrate the effectiveness of our gradient-based masking strategy, we compare LeaF with two other choices of token masking: (1) **Random Masking**, where the same number of tokens identified by our gradient-based method are randomly masked for each instance; and (2) **PPL-based Masking**, where tokens with the highest perplexity scores are masked at the same ratio, following the same data filtering process used in our method to generate augmented training data. We evaluate all three masking strategies, Random Masking, PPL-based Masking, and Gradient-based Masking, on the GSM8K, MATH-500, and OlympiadBench datasets.

**Results.** We present the results in Figure 7. Our observations are as follows: (1) Gradient-based Masking (ours) consistently outperforms both baselines, with the highest accuracy on MATH-500 and OlympiadBench. (2) Random Masking leads to performance degradation on GSM8K and Olympiad, despite showing a slight improvement on MATH-500. This suggests that naively masking tokens without informed selection can undermine distillation performance. (3) PPL-based masking provides modest improvements on GSM8K and MATH-500, but performs comparably to random masking on OlympiadBench, indicating its limitations on complex tasks. Our analysis is, while perplexity may suffice for detecting confounding tokens in simpler settings, it lacks the sensitivity needed for challenging benchmarks. This highlights the necessity of an advanced teacher model to guide token selection in complex reasoning scenarios.

## 4.2 Response Splitting Strategies

We compare three response splitting strategies, considering that distracting patterns exist at both the instruction and response levels. We evaluate response without split (instruction-level pruning), 2-segment splits, and 3-segment splits. A diagram illustrating these strategies is shown in Figure 6.

**Results.** As shown in the results in Figure 8, our findings are as follows: (1) Response-level pruning (both 2-segment and 3-segment splits) significantly outperforms instruct-level pruning, demonstrating the importance and benefits of extending the learning of confounding tokens from the instruction level to the response level. We hypothesize that the improvement stems from differences in distracting patterns between the instruction and response levels, suggesting that combining both levels could further enhance model performance. (2) The performance of 3-segment splits is comparable to that of 2-segment splits, suggesting that further segmentation at the response level yields diminishing returns. We hypothesize that distracting patterns at the response level exhibit certain regularities, and the data generated by 2-segment splits is sufficient for the model to learn these patterns effectively, rendering additional segmentation unnecessary.

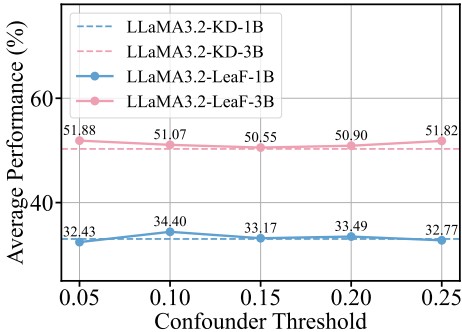

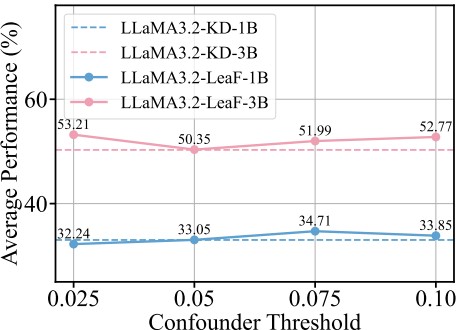

Figure 9: Instruct-level in MathBench.     Figure 10: Response-level in MathBench.

### 4.3 Threshold Sensitivity Analysis

We perform a sensitivity analysis on the threshold used for confounding tokens pruning, assessing the performance of LLaMA3.2-1B-Instruct and LLaMA3.2-3B-Instruct as student models. As illustrated in the Figure 9 and Figure 10, we present the average experimental results of these models across MathBench (GSM8K, MATH-500, OlympiadBench) at different threshold levels.

**Results.** We observe the following results: (1) Instruct-level: LLaMa3.2-LeaF-1B performs best at a threshold of 0.10, while LLaMa3.2-LeaF-3B performs best at 0.05. (2) Response-level: LLaMa3.2-LeaF-1B performs best at 0.15, and LLaMa3.2-LeaF-3B at 0.10. (3) For both instruction-level and response-level, LLaMa3.2-LeaF-1B achieves optimal performance at a higher misleading token threshold than LLaMa3.2-LeaF-3B. This suggests that smaller models, which are more susceptible to confounding tokens, benefit from higher thresholds that filter out disruptive tokens more effectively. Detailed cross-domain results presented in Appendix G further confirm that $\tau_{\mathrm{mis}} = 0.10$ yields stable performance across tasks.

### 4.4 Case Study in an Interpretable Perspective

We conduct a case study from an interpretability perspective to show that LeaF, compared to standard knowledge distillation (KD), enables the model to focus on critical information and avoid confounding tokens. The interpretability case study is shown in Figure 11. It illustrates that LeaF allows the model to focus more on the critical information, such as "real number", "all", "are real". By identifying the requirement that all roots be real, LeaF guides the model through a coherent reasoning chain: recognizing $x = -1$ as an evident real root, and applying the discriminant condition to ensure the quadratic factor also yields real solutions. In contrast, the KD model overlooks this constraint and misapplies the AM–GM inequality to potentially negative values, resulting in an incorrect final answer. This demonstrates that LeaF enables the model to focus on critical information better.

## 5 Related Work

**Chain-of-Thought Knowledge Distillation.** Our work falls within the knowledge distillation paradigm [18, 49], a widely adopted technique for boosting the performance of open-source models in complex tasks such as mathematical problem-solving [58, 59] and code generation [3, 21]. CoT-KD [41] takes the first step towards this line to employ chain-of-thought explanations generated by a high-capacity teacher to fine-tune a student model, thereby equipping it with advanced reasoning capabilities. Recent advancements can be broadly categorized into two complementary directions: (1) **Data-focused approaches** aim to enhance distillation by improving the quality and diversity of training data (e.g., CD [13], SCORE [56], and Skintern [28]). (2) **Model-focused approaches** enhance model architectures and inference strategies to improve efficiency and reasoning capability (e.g., PRR [41] and ATM [7]). However, existing methods primarily focus on output imitation, while our approach explicitly distills the teacher's ability to capture critical information during inference into the student, enabling context-aware reasoning in the student model.

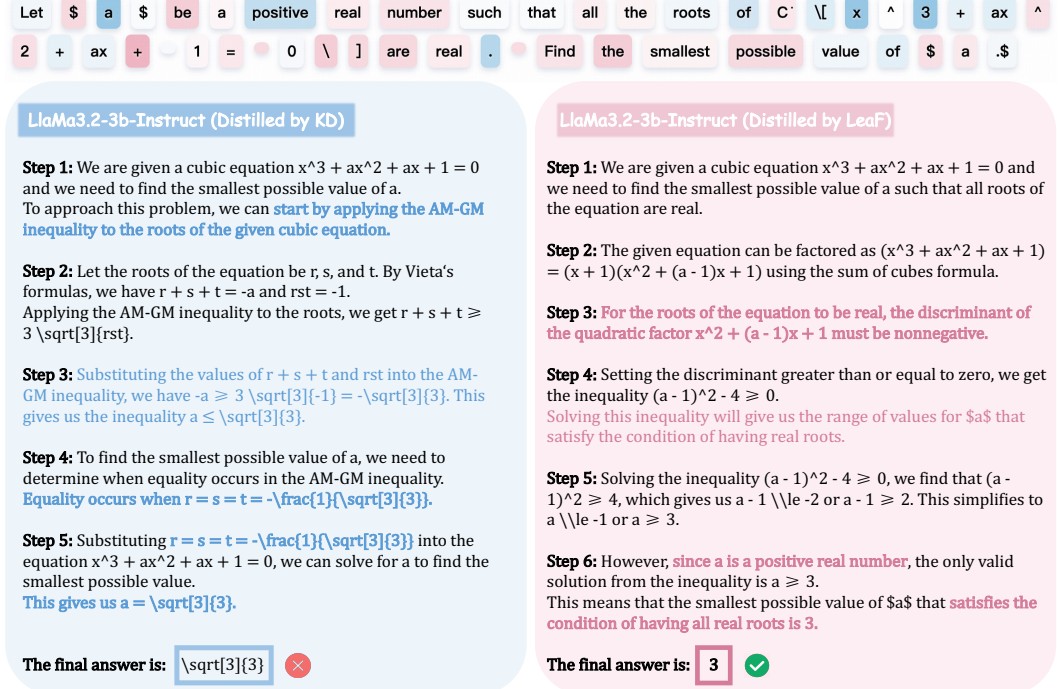

Figure 11: Case study comparing LeaF and knowledge distillation (KD) performance on the MATH-500. **Top:** Heatmap showing the attention score differences between KD and LeaF on instruction tokens. Dark blue indicates higher KD attention, and deep pink indicates higher LeaF attention. **Left:** Response of LLaMA-3.2-3B-Instruct after standard knowledge distillation, with blue text marking incorrect steps by the KD model. **Right:** Response of LLaMA-3.2-LeaF-3B, with pink text highlighting correct steps by our model.

**Critic Token Identification.** Existing works have explored various strategies to identify and mitigate redundant steps [10, 33, 35] or less-important tokens [15, 47] during language model reasoning. Methods like LLMLingua [23] rely on models' self-assessment to determine token importance, which may introduce biases due to the model's limited reasoning capacity. More recent approaches further refine token selection across different training stages. RHO-1 [29] introduces Selective Language Modeling (SLM) during the Supervised Fine-Tuning phase, prioritizing informative tokens to enhance efficiency and reasoning accuracy. TokenSkip [47] focuses on Chain-of-Thought stages, selectively skipping low-impact tokens to compress reasoning paths without sacrificing performance. Meanwhile, cDPO [30] enhances Direct Preference Optimization by isolating critical tokens through contrastive learning. However, these methods primarily concentrate on output tokens while overlooking the influence of prior contextual information from the instruction phase. In contrast, our method introduces an advanced teacher model to identify confounding tokens based on gradient differences, establishing stronger connections between instruction and output, and achieving more holistic and effective token filtering.

**Reasoning Consistency.** A detailed discussion of reasoning consistency is provided in Appendix I.

## 6    Conclusion

In this paper, we proposed Learning to Focus Framework (LeaF), a novel strategy to enhance the consistency of large language models. By leveraging causal analysis and gradient-based pruning, our method effectively identifies and eliminates confounding tokens, enabling the student model to capture more reliable causal dependencies. Experimental results show substantial improvements in both accuracy and robustness across mathematical reasoning, code generation and multi-hop QA benchmarks. In future work, how to achieve model consistency through self-improvement mechanisms without relying on advanced models remains a promising direction for further exploration.

## Acknowledgements

This work was supported by The National Natural Science Foundation of China (No. 62376273 and No.U2436209) and Beijing Nova Program (No. 20240484568).

We sincerely thank all the anonymous reviewers and (S)ACs for their constructive comments, as well as Zhen Tian, Tingchen Fu, Wentong Chen, Yongda Yu, Hongjia Liu and Qinghui Wang for their valuable discussions on both the development of the ideas and the writing of this paper.

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

# A    Preliminary Experiment

## A.1    Gradient Heatmap Comparison

We compare the gradient heatmaps of the small model and the big model to analyze the differences in their attention and focus on critical tokens during the inference process.

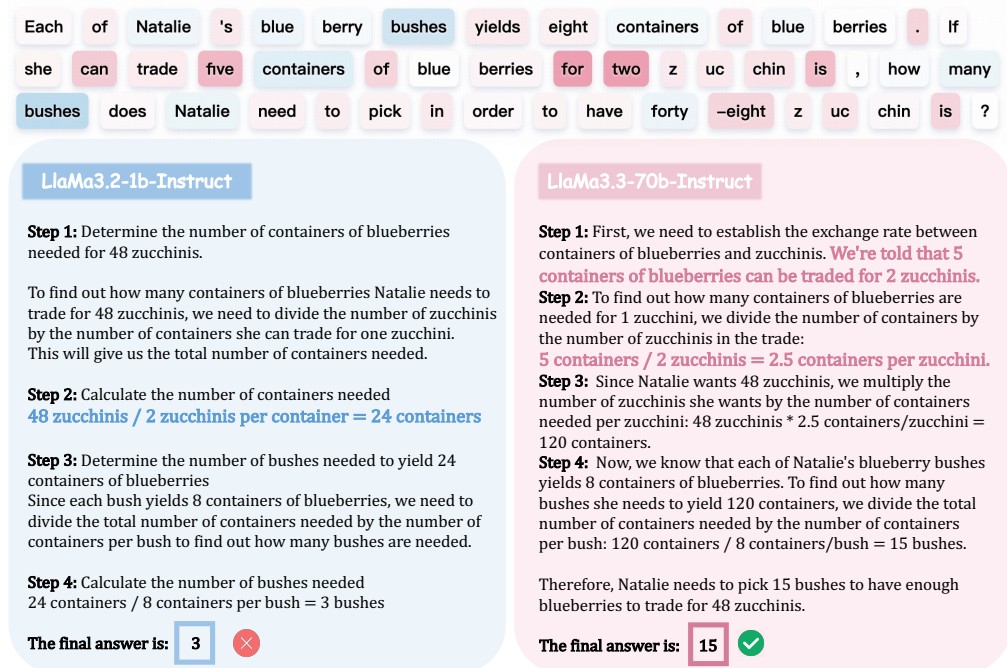

Figure 12: A case study comparing the performance of the student model (LlaMa3.2-1b-Instruct) and the teacher model (LlaMa3.3-70b-Instruct) on the MATH task.

**Results.**    Figure 12 illustrates that Llama3.3-70B-Instruct successfully captures a key contextual relation: "Five containers of blueberries can be traded for two zucchinis." Gradient heatmaps show that the teacher model aligns its attention closely with the relevant tokens, while the student model's attention is more dispersed. This observation motivates our hypothesis: **by pruning distracting patterns, we can guide the student model to better focus on salient information, enhancing its reasoning capabilities.** To evaluate this, we conduct pilot studies in two settings: (1) assessing accuracy gains on math and code benchmarks after pruning distracting patterns (Appendix A.2) and (2) evaluating improvements in response quality(Appendix A.3).

## A.2    Performance Gains from Token Pruning

For the LLaMA series, we use LLaMA3.2-1B/3B-Instruct as the student models and LLaMA3.3-70B-Instruct as the teacher model. For the Qwen series, we use Qwen2.5-Math-1.5B as the student model and Qwen2.5-72B-Instruct as the teacher model for preliminary experiments.

For mathematical problem solving, we select a filtered subset from the Numina-CoT dataset [26], where the teacher models generate correct inferences, consisting of 12K examples (3K each from GSM8K, Olympiads, AMC_AIME, and MATH). We then select the subset where the student models produce incorrect results. We compute each sample's gradient difference between the student and teacher models to identify potential distracting patterns. Finally, these distracting patterns are removed, and the student models are re-evaluated to assess the resulting improvement in accuracy. The results are presented in Figure 1(b).

For code generation, we construct a filtered subset of 12K examples from the AceCode dataset [53], where teacher models generate correct inferences. This subset comprises 6K samples from the Evol

subset and 6K from the Oss subset. We apply the same distracting pattern identification and pruning procedure as used in the mathematical domain. Subsequently, we re-evaluate student models to assess the impact of this intervention on code generation accuracy. The results are presented in Figure 1(a).

### A.3 Generation Quality Improvements from Token Pruning

We analyze the alignment between the student model's outputs and the teacher model's outputs under two conditions: (1) **Original Instruction**: The student model generates output based on the original

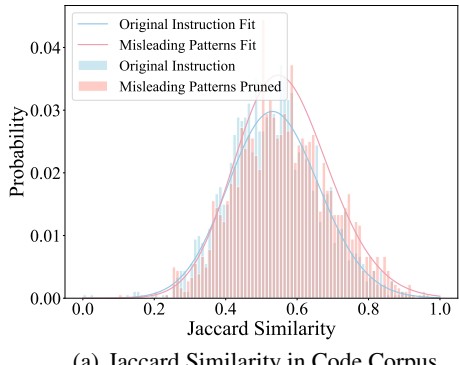
(a) Jaccard Similarity in Code Corpus

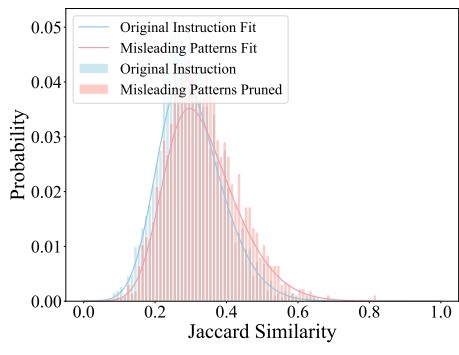
(b) Jaccard Similarity in Math Corpus

Figure 13: Jaccard similarity distribution between student model responses (original vs. instruction pruned distracting patterns) and ground-truth responses on math and code datasets.

instruction. (2) **Instruction pruned distracting patterns**: The student model generates output from an instruction where confounding tokens are masked. To quantify the similarity between the model outputs, we use two widely adopted metrics: Jaccard Similarity [54]. This metric enables a practical evaluation of how well the student model's output aligns with the contextual meaning conveyed by the teacher model.

**Results.** Figure 13 show a shift in Jaccard Similarity distribution for responses generated by the student model on both code and math tasks after removing confounding tokens. This indicates that, by ignoring distracting patterns, the student model not only improves reasoning accuracy (Figure 1) but also generates responses more aligned with the teacher model, thereby enhancing output quality.

## B Detailed Evaluation Settings

For mathematical problem solving, we evaluate on GSM8K, MATH, and OlympiadBench using the Step-DPO framework [24], with modifications to address its data extraction inaccuracies. For code generation, we report pass@1 on HumanEval(+) [6] and LeetCode [9], using EvalPlus [32], and pass@10 on LiveCodeBench [22], using the Skythought-Evals framework[25]. For multi-hop question answering, we evaluate on HotpotQA [50], 2WikiMultiHopQA [19], and Musique [44] using the LongMab-PO framework [12], with all datasets sourced from LongBench [5].

## C Collective Masking vs. Span Masking

We compare two pruning strategies (Figure 5): **Collective Pruning**, which simultaneously masks all identified confounding tokens, and **Span Pruning**, which masks only contiguous spans of tokens.

**Results.** Table 3 presents results for LLaMA3.2-1B/3B-Instruct models trained on Math corpus (86K) and evaluated on MATH-500 under two pruning schemes. (1) We find that span pruning substantially outperforms both collective Pruning and native knowledge distillation. (2) Collective Pruning not only fails to yield improvements but even degrades performance on the LLaMA3.2-3B-Instruct model. We suspect that pruning all confounding tokens at once disrupts the training data's

semantic coherence, undermining the model's learning. Consequently, we adopt the **Span Pruning strategy** for all subsequent experiments.

Table 3: Comparison of two pruning strategies: Collective Pruning vs. Span Pruning on MATH-500 for LLaMA3.2–1B/3B-Instruct. The best and second-best results are marked in **bold** and underlined, respectively.

| Model | MATH-500 |
|---|---|
| *LLaMA3.2-1B-Instruct* | |
| Instruct Model (Pre-KD) | 24.20 |
| KD w/o Mask | 34.00 |
| Collective Pruning | 34.20 |
| Span Pruning | **37.40** |
| *LLaMA3.2-3B-Instruct* | |
| Instruct Model (Pre-KD) | 42.80 |
| KD w/o Mask | 50.00 |
| Collective Pruning | 49.20 |
| Span Pruning | **54.40** |

# D   Computational Overhead Analysis

We report detailed runtime and memory analyses of LeaF compared with standard knowledge distillation (KD) under identical hardware configurations.

**Gradient Computation.**   Gradient difference computation in LeaF is a one-time offline process on 8×NVIDIA A100 (80GB) GPUs, jointly computing gradients from the teacher and student models. Processing around 7K samples takes about 3 hours.

**Counterfactual Generation.**   Counterfactual responses are generated offline using vLLM on 4×NVIDIA A100 (80GB) GPUs. Processing 26K samples requires approximately 50 minutes for smaller models (e.g., LLaMA-3.2-1B-Instruct) and up to 2.85 hours for larger ones (e.g., LLaMA-3.3-70B-Instruct or Qwen2.5-72B-Instruct). This step is parallelizable and executed only once.

**Training Overhead.**   We further measure the end-to-end training overhead over 3 epochs on 4×NVIDIA A100 (80GB) GPUs. Results are reported in Table 4. LeaF incurs an additional 10–13% training time compared to standard KD.

Table 4: End-to-end runtime comparison between standard KD and LeaF.

| Model | KD Runtime (h) | LeaF Runtime (h) | Overhead (%) |
|---|---|---|---|
| LLaMA-3.2-1B-Instruct | 26.2 | 29.2 | +11.5 |
| LLaMA-3.2-3B-Instruct | 23.7 | 26.2 | +10.6 |
| Qwen2.5-Math-1.5B | 27.3 | 30.9 | +13.3 |

**Results.**   LeaF introduces a moderate training overhead of 10–13% compared to standard KD, yet consistently yields 2–3% absolute accuracy gains on mathematics, coding, and multi-hop QA benchmarks. This improvement indicates that the additional computation is effectively utilized to enhance reasoning performance. Its auxiliary procedures, including gradient normalization, span pruning, and counterfactual generation, are one-time, offline, and fully parallelizable, incurring no extra cost during training or inference. These properties make LeaF practical and scalable for large-scale applications.

# E    Robustness Analysis

To evaluate test-time robustness, we assess LeaF on perturbed versions of the MathBench benchmarks (GSM8K, MATH-500, and OlympiadBench). Perturbations are generated through back-translation, a standard data augmentation technique for producing realistic paraphrastic variations.

Table 5: Performance on perturbed MathBench datasets for Instruct models (LLaMA-3.2-1B/3B-Instruct) across different pruning strategies.

| Model | GSM8K | MATH-500 | OlympiadBench | Avg. |
|---|---|---|---|---|
| *LLaMA 3.2–1B-Instruct* | | | | |
| Instruct (Pre-KD) | 39.65 | 24.80 | 3.86 | 22.77 |
| KD (no mask) | 50.42 | 31.80 | 5.19 | 29.14 |
| LeaF (Instr Mask) | 51.10 | 32.00 | 6.53 | 29.88 |
| LeaF (Instr & Resp Mask) | **51.71** | **34.20** | **6.97** | **30.96** |
| *LLaMA 3.2–3B-Instruct* | | | | |
| Instruct (Pre-KD) | 70.43 | 40.80 | 9.64 | 40.29 |
| KD (no mask) | 74.00 | 48.20 | 14.09 | 45.43 |
| LeaF (Instr Mask) | 74.07 | 50.20 | 15.58 | 46.62 |
| LeaF (Instr & Resp Mask) | **76.12** | **51.20** | **19.88** | **49.07** |

**Results.**    Table 5 reports the results for LLaMA-3.2-1B/3B-Instruct models evaluated on perturbed MathBench datasets. (1) LeaF consistently outperforms standard KD across noisy conditions, indicating that its pruning and attribution mechanisms are robust to moderate linguistic perturbations. (2) The performance degradation under noise remains within 2–3%, suggesting that LeaF effectively preserves reasoning capability even when input distributions shift. These findings highlight LeaF's robustness for real-world scenarios involving input perturbations or distributional noise.

# F    Ablation Study: Importance of Contrastive Pairs

We include an ablation setting (*w/o Student-Wrong Originals*) to examine the impact of excluding original samples that the student model fails to solve due to confounding tokens. In contrast, LeaF leverages both student-correct and student-wrong originals together with counterfactual samples.

Table 6: Ablation results on MathBench comparing LeaF and *w/o Student-Wrong Originals*. The ablated variant uses only student-correct originals and counterfactuals, excluding student-wrong ones.

| Model | GSM8K | MATH-500 | OlympiadBench | Avg. |
|---|---|---|---|---|
| *LLaMA 3.2–1B-Instruct* | | | | |
| LeaF | **57.70** | **35.40** | **10.09** | **34.40** |
| w/o Student-Wrong Originals | 58.15 | 34.80 | 7.42 | 33.46 |
| *LLaMA 3.2–3B-Instruct* | | | | |
| LeaF | **83.09** | **51.80** | **20.77** | **51.88** |
| w/o Student-Wrong Originals | 84.08 | 47.80 | 16.02 | 49.30 |

**Results.**    (1) LeaF outperforms the ablated variant on MATH-500 and OlympiadBench, demonstrating that retaining original samples containing misleading patterns alongside counterfactual samples provides a stronger contrastive signal. This helps the student model downweight spurious correlations and attend to causally relevant information. (2) Excluding student-wrong originals reduces exposure to misleading cases, limiting the model's ability to generalize under confounding conditions. (3) The slight performance gain of the ablated variant on GSM8K stems from dataset composition. Compared with MATH-500 and OlympiadBench, GSM8K is a simpler dataset, and after filtering, its student-correct samples constitute a larger share of the training data. This distributional bias favors easier problems, leading to higher accuracy on GSM8K.

## G  Threshold Stability Across Tasks

We further examine the stability of the threshold ($\tau_{\text{mis}}$) across domains, including math (*GSM8K*, *MATH* and *OlympiadBench*), code (*HumanEval+*, *LeetCode* and *LivecodeBench(V4)*), and question answering (*SQuAD 2.0*). For each task, model accuracy is evaluated at three threshold settings ($\tau_{\text{mis}} = 0.05$, 0.10, and 0.15) using *LLaMA-1B-Instruct* and *LLaMA-3B-Instruct* as student models. Table 7 summarizes the results.

Table 7: Accuracy under different $\tau_{\text{mis}}$ thresholds across tasks.

| Task | Model | $\tau$=0.05 | $\tau$=0.10 | $\tau$=0.15 |
|------|-------|-------------|-------------|-------------|
| MathBench | LLaMA-3.2-1B-Instruct | 32.43 | **34.40** | 33.17 |
| | LLaMA-3.2-3B-Instruct | **51.88** | 51.07 | 50.55 |
| CodeBench | LLaMA-3.2-1B-Instruct | 17.85 | 18.27 | **19.76** |
| | LLaMA-3.2-3B-Instruct | 32.83 | **33.55** | 32.95 |
| SQuAD 2.0 | LLaMA-3.2-1B-Instruct | 72.12 | 73.33 | **74.78** |
| | LLaMA-3.2-3B-Instruct | 88.67 | **89.44** | 83.66 |

**Results.**  (1) $\tau_{\text{mis}} = 0.10$ yields the best or near-best performance, indicating that this threshold value is robust across domains. (2) Smaller models (*LLaMA-1B-Instruct*) exhibit mild sensitivity to larger thresholds, while larger models (*LLaMA-3B-Instruct*) remain stable around $\tau_{\text{mis}} = 0.10$. (3) These findings indicate that $\tau_{\text{mis}} = 0.10$ achieves a stable trade-off between filtering noisy gradients and preserving useful learning signals.

## H  Statistics of Counterfactual and Original Samples

We report token-length statistics and sample counts for the original and counterfactual datasets used in training. Results are presented separately for the math and code tasks, covering both LLaMA-3.2-1B-Instruct and LLaMA-3.2-3B-Instruct. For each subset, we include the minimum, maximum, and average token lengths, along with total sample counts and subtask-level breakdowns.

Table 8: Statistics of original and CF (counterfactual) samples for the math task.

| Model | Min | Max | Avg. | Total | AIME | MATH | GSM8K | OlympiadBench |
|-------|-----|-----|------|-------|------|------|-------|---------------|
| Original Samples | 23 | 2711 | 98.50 | 12000 | 3000 | 3000 | 3000 | 3000 |
| LLaMA-3.2-1B CF | 24 | 2710 | 160.78 | 4576 | 1620 | 1029 | 907 | 1020 |
| LLaMA-3.2-3B CF | 23 | 2711 | 148.79 | 2843 | 1228 | 709 | 172 | 734 |

Table 9: Statistics of original and CF (counterfactual) samples for the code task.

| Model | Min | Max | Avg. | Total |
|-------|-----|-----|------|-------|
| Original Samples | 34 | 527 | 140.17 | 12000 |
| LLaMA-3.2-1B CF | 47 | 507 | 158.61 | 6210 |
| LLaMA-3.2-3B CF | 46 | 446 | 159.72 | 4059 |

**Results.**  Table 8 and Table 9 summarize the distributional properties of the training data used in the main experiments. We observe that the average token length of counterfactual samples is slightly higher than that of the original samples in both the math and code tasks. This pattern likely arises because longer and more complex problems tend to include misleading patterns for student models, increasing the likelihood of counterfactual generation.

# I  Extended Related Work

**Reasoning Consistency.** A series of existing works have focused on decoding-phase consistency. Self-Consistency [46] stabilizes final answers by sampling multiple reasoning chains and applying majority voting. To further reduce inference costs, Adaptive Consistency [2] and Early Scoring Self-Consistency [27] introduce Dirichlet-based and window-based stopping criteria, respectively. Reasoning Aware Self-Consistency [45] enhances answer consistency by weighting sample quality and reasoning-path importance. However, these methods primarily focus on stabilizing the final answer without addressing the model's ability to maintain and leverage key contextual information throughout the generation process. We instead define consistency as both answer stability and context adherence, and introduce a distillation strategy that systematically suppresses the influence of misleading signals, enhancing the student's focus on key contextual information and promoting more robust context adherence during inference.

# J  Detailed Training Settings

The complete training hyper-parameters in knowledge distillation are put in Table 10.

Table 10: Training hyper-parameters in Knowledge Distillation.

| Model | Hyper-parameter | Value |
|---|---|---|
| LLaMA3.2-1B-Instruct | LR | $1 \times 10^{-5}$ |
| | LR Scheduler | cosine |
| | Batch Size | 64 |
| | Epochs | 3 |
| | Maximum Sequence Length | 4096 |
| | Warmup Steps | 5 |
| | Distill Loss Type | KL |
| | Validation Set Size (Math) | 1035 |
| | Validation Set Size (Code) | 2000 |
| LLaMA3.2-3B-Instruct | LR | $1 \times 10^{-5}$ |
| | LR Scheduler | cosine |
| | Batch Size | 64 |
| | Epochs | 3 |
| | Maximum Sequence Length | 3000 |
| | Warmup Steps | 5 |
| | Distill Loss Type | KL |
| | Validation Set Size (Math) | 1035 |
| | Validation Set Size (Code) | 2000 |
| Qwen2.5-Math-1.5B | LR | $1 \times 10^{-5}$ |
| | LR Scheduler | cosine |
| | Batch Size | 32 |
| | Epochs | 3 |
| | Maximum Sequence Length | 4096 |
| | Warmup Steps | 5 |
| | Distill Loss Type | KL |
| | Validation Set Size (Math) | 1200 |
| | Validation Set Size (Code) | 2000 |

# K  Evaluation Benchmarks

GSM8K [8] comprises 8500 grade-school–level word problems, each requiring 2–8 steps of basic arithmetic. Its natural-language diversity and multi-step structure make it a standard measure for chain-of-thought prompting.

MATH [17] contains 12500 competition-style problems grouped into seven topics (Prealgebra, Algebra, Number Theory, Counting & Probability, Geometry, Intermediate Algebra, Precalculus). Every question is accompanied by a detailed solution.

OlympiadBench [16] was originally a bilingual, multimodal collection of 8476 Olympiad-level math and physics problems. We filter out proof-based and image-based questions to obtain 674 pure-text tasks, enabling focused evaluation of advanced symbolic reasoning.

HumanEval+ [32] extends the original HumanEval with additional Python programming tasks and augmented unit tests, targeting functional correctness across diverse code patterns.

LeetCode [9] samples real-world algorithmic challenges from the LeetCode platform—arrays, trees, dynamic programming, etc.—to assess models' ability to generate correct and efficient solutions.

LiveCodeBench [22] provides a large-scale suite of real-world coding tasks with comprehensive unit tests and human preference annotations, allowing evaluation of both functional accuracy and coding style.

HotpotQA [50] consists of question–answer pairs that require reasoning over multiple Wikipedia paragraphs. It evaluates models' ability to retrieve and combine evidence from different sources to answer complex questions.

2WikiMultiHopQA [19] includes multi-hop questions automatically generated from Wikipedia, where each question involves two entities and demands reasoning across at least two documents.

Musique [44] provides multi-hop questions with fine-grained decomposition and supporting evidence. It is designed to test compositional reasoning and factual consistency across multiple textual contexts.

## L  Open-source Instruct Models

The download links for the four open-source models are provided below:

- Llama-3.2-1B-Instruct: https://huggingface.co/meta-llama/Llama-3.2-1B-Instruct
- Llama-3.2-3B-Instruct: https://huggingface.co/meta-llama/Llama-3.2-3B-Instruct
- Llama-3.3-70B-Instruct: https://huggingface.co/meta-llama/Llama-3.3-70B-Instruct
- Qwen2.5-Math-1.5B: https://huggingface.co/Qwen/Qwen2.5-Math-1.5B
- Qwen2.5-72B-Instruct: https://huggingface.co/Qwen/Qwen2.5-72B-Instruct

## M  Limitations

Our work has the following limitations: (1) **Dependence on an advanced teacher model**: Our method relies on a teacher model to identify confounding tokens. Exploring self-improvement that enables models to refine their attention to critical tokens and boost reasoning can be an interesting and important future work. (2) **Limited scalability to long-form generation**: Due to the inherent limitations of the student model, we have only validated our approach on math and code tasks, leaving its applicability to long-text generation and other domains for future investigation.

