# OpenReview forum: "Learning to Focus: Causal Attention Distillation via Gradient‐Guided Token Pruning"
_NeurIPS.cc/2025/Conference — NeurIPS 2025 poster_

### Official Review · Reviewer_WSL6 · 2025-07-02

**Clarity:** 2
**Significance:** 4
**Originality:** 3
**Rating:** 4
**Confidence:** 3

**Summary:**

This paper proposes a knowledge distillation methodology that detects misleading tokens in inputs and outputs during LLM reasoning and makes the model robust to them. The authors argue that LLMs have tokens with spurious correlations unrelated to actual correct answer generation, which hinder accurate and consistent response generation during reasoning processes. To address this, the study defines misleading tokens as those with small gradient magnitude differences between teacher-student models, and performs knowledge distillation using augmented datasets that exclude these tokens. Through various experimental results, they demonstrate that this masking-distillation strategy improves the model's reasoning performance and show that the detected tokens indeed need to be removed for performance improvement.

**Questions:**

1. A detailed explanation of Token masking and Data Augmentation in Section 2.2 seems necessary. Misleading tokens appear to be detected independently, but looking at Section 4.2 experiments, it seems like consecutive spans are masked. How is the process of converting individual masking tokens to consecutive spans handled?
2. From my understanding, misleading token masking is performed on consecutive spans. In Subsection 4.1, is random masking performed on arbitrary consecutive tokens? Or on independent tokens?
3. By what criteria were misleading tokens identified in the preliminary experiment in Figure 1? Did this experiment use the same method as the proposed methodology's gradient-based approach?
4. How is the threshold $\tau_\text{mis}$ in Equation 6 determined? While sensitivity analysis results are said to be in the Appendix, they're currently difficult to verify and more detailed explanation in the main text is needed.
5. What is the process of defining misleading tokens as those with small gradient magnitude differences between teacher-student in Equations 5 and 6? An intuitive explanation seems necessary.

**Ethical Concerns:**

["NO or VERY MINOR ethics concerns only"]

**Final Justification:**

After carefully considering the authors’ rebuttal and additional clarifications, I have decided to raise my score from 2 (Reject) to 4 (Borderline Accept). The authors have addressed most of my initial concerns regarding the technical soundness and experimental validation. In particular, they provided:
- Additional experiments and ablation studies that strengthen the empirical evidence for the proposed method.
- Clarifications on the design choices and assumptions, which resolve the ambiguities I previously noted.
- More detailed description of the dataset composition and experimental setup, improving the reproducibility of the work.

While certain aspects, such as the rigor of the causal framework, could still be improved, the overall contribution is now technically solid, and the reasons to accept outweigh those to reject. Therefore, I consider the paper a borderline accept.

**Limitations:**

yes

**Paper Formatting Concerns:**

1. [line 58] "Casual Attention Distillation" → "Causal Attention Distillation"
2. The appendix is not attached to the paper, making verification difficult.

**Quality:**

2

**Strengths And Weaknesses:**

### Strengths:
1. Addresses a frequently occurring phenomenon directly. Reasoning models indeed show phenomena where they incorrectly focus on very trivial elements during answer generation, and weaker models especially fail to properly exploit reasoning paths by focusing on wrong patterns. The motivation of this research is very intuitive regarding this issue, and resolving spurious correlations appears important.
2. Attempted to interpret spurious tokens through a causal framework, trying to capture relationships not only in instructions but also within reasoning steps.
3. Shows consistent performance improvements. As confirmed through Table 1, performance consistently improves across different model types and sizes compared to baselines.

### Weaknesses:

1. Lack of validity in core assumptions: While Table 1 shows that the proposed methodology brings consistent performance improvements, it's difficult to clearly distinguish whether this is truly due to "misleading" tokens or other factors such as sequence length reduction, noise reduction, or data augmentation.
2. Limitations in misleading token identification method: There's insufficient theoretical/empirical evidence that gradient-based comparison (Equation 5) truly finds "misleading" tokens. Equations 5 and 6 appear to simply select tokens with small gradient differences between teacher-student models, and it's difficult to find experiments and evidence in the paper showing that these tokens have spurious correlations.
3. Superficial application of causal framework: While citing Pearl's causal framework, it essentially amounts to simple token masking. Particularly, as seen in Figure 2's example, tokens detected as misleading actually appear to be necessary information for answer generation. This makes it difficult to argue that these tokens have spurious correlations and that masking resolves this.
4. Weak baselines: Random masking baseline is too simple, and comparison with more advanced distillation methods is lacking. Direct comparison with CoT knowledge distillation methodologies mentioned in the paper's related works seems necessary. This could demonstrate that it's not simply about distilling CoT reasoning well, but that measures against spurious correlations are needed.

---

> ### Author Rebuttal · Authors · 2025-07-31
>
> **Q1: Lack of validity in core assumptions: While Table 1 shows that the proposed methodology brings consistent performance improvements, it's difficult to clearly distinguish whether this is truly due to "misleading" tokens or other factors such as sequence length reduction, noise reduction, or data augmentation.**
>
> **A1:** We address the concern by explicitly isolating the impact of misleading tokens from other confounding factors:
> First, we rule out **sequence length reduction** and **data augmentation** effects. In Section 4.2 (Random Masking setting), we applied masking with exactly the same proportion as CAD, ensuring identical amounts of sequence length reduction and data augmentation across methods. The results show that random masking provides no improvement—and in fact slightly degrades performance on GSM8K and Olympiad tasks compared to standard distillation—confirming that CAD’s gains cannot be attributed simply to reduced sequence lengths or increased data diversity.
> |                | GSM8K |  MATH | OlympiadBench |    Avg.   |
> |----------------|:-----:|:-----:|:-------------:|:---------:|
> | KD w/o Mask    | 82.87 | 49.00 |     18.99     |   50.29   |
> | Random Masking | 82.41 | 51.00 |   **14.54**   |   49.32   |
> | CAD            | 83.09 | 51.80 |   **20.77**   | **51.88** |
>
> Second, we eliminate the confounding benefits from **distillation with large models and chain-of-thought (CoT)** by comparing our approach directly against a strong CoT-KD baseline (Table 1).
> CAD consistently delivers additional improvements over CoT-KD on both Math and Code tasks, highlighting that our approach provides value beyond these established techniques.
>
> Finally, we validate that CAD’s improved reasoning ability arises from better focus on critical information through interpretability analysis (In Section 4.3). As shown in Figure 8, CAD allows the model to attend more effectively to key problem elements such as “multiple” and “5” (e.g., combinations like (1,5), (2,5)), which directly relate to the condition “product of two numbers is a multiple of 5.” In contrast, the KD baseline fails to sufficiently focus on these tokens and instead considers invalid pairs (e.g., 6 and 3), leading to incorrect final answers.
>
> **Q2: What is the process of defining misleading tokens as those with small gradient magnitude differences between teacher-student in Equations 5 and 6? An intuitive explanation seems necessary.**
>
> **A2:** We appreciate the opportunity to clarify our method of defining misleading tokens, as there seems to be a misunderstanding.
> We do **not simply select tokens with small gradient magnitude differences** between teacher and student models. Instead, our approach specifically targets tokens on which the **student model excessively relies** (high gradient magnitude) but which the **teacher model largely ignores** (low gradient magnitude), particularly on samples where the student makes an incorrect prediction while the teacher correctly solves the task.
>
> More concretely, we first select examples for which the student model is incorrect and the teacher model correct. Next, we employ Equations 5 and 6 to compute gradients with respect to tokens, and identify tokens exhibiting a significant discrepancy—namely, large student gradients versus small teacher gradients—as candidate misleading tokens.
>
> Crucially, after removing these candidate tokens, the student’s predictions often change from incorrect to correct, while the teacher’s correct reasoning remains unaffected. This indicates that the student initially paid undue attention to these tokens at the expense of genuinely informative tokens. Consequently, we designate these tokens as **misleading tokens** and use them to generate counterfactual training samples, guiding the student model to better align its attention with truly relevant information.
>
> **Q3(a)：Superficial application of causal framework: While citing Pearl's causal framework, it essentially amounts to simple token masking.**
>
> **A3(a):** We appreciate the opportunity to clarify why CAD represents more than simple token masking and how it aligns with Pearl’s causal framework.
>
> Our preliminary experiments showed that directly removing certain "misleading" tokens could flip a model’s prediction from incorrect to correct—even **without retraining**. This aligns closely with Pearl’s causal perspective: a subset of input tokens ($A$) can act as confounders, creating a spurious correlation ($A \to Y$). By explicitly performing a causal intervention—masking these misleading tokens—we break the spurious pathway, restoring the true causal relationship ($X \to Y$). CAD thus leverages these carefully constructed counterfactual examples during distillation, teaching models to ignore spurious signals.
> Thus, CAD is not merely token masking—it is a principled causal intervention designed explicitly to eliminate spurious correlations while preserving essential information.
>
> **Q3(b)：Particularly, as seen in Figure 2's example, tokens detected as misleading actually appear to be necessary information for answer generation. This makes it difficult to argue that these tokens have spurious correlations and that masking resolves this.**
>
> **A3(b):** While the tokens identified as misleading (e.g., in Figure 2) may appear necessary at first glance, the information they carry is often redundant and can be inferred from other parts of the prompt.
> In Figure 2, the phrase **“the number of bacteria doubles every day”** is marked as misleading patterns. Although it appears informative, this detail is already implied by the earlier sentence: **“The colony starts with 3 bacteria, and has 6 at the end of day 1.”** From this, it is evident that the bacteria count doubles daily. Therefore, explicitly stating the doubling rule is not strictly necessary for reasoning.
> The issue arises when the model over-attends to such redundant information. In this example, excessive focus on the doubling phrase may lead the model to overlook truly critical conditions, such as **“ends with the colony having more than 100 bacteria.”** This misalignment in attention can cause the model to produce incorrect answers, despite having access to all necessary information.
> By training on counterfactual examples where only the misleading tokens are masked, CAD encourages models to redistribute their attention across all genuinely informative tokens, thereby improving overall reasoning robustness.
>
> **Q4：Weak baselines: The random masking baseline is too simple, and comparisons to advanced methods like CoT knowledge distillation are missing. A direct comparison is needed to show that improvements come from addressing spurious correlations, not just better CoT distillation.**
>
> **A4:** The **KD w/o Mask baselines** reported in Table 1 of Section 3.2 (including LLaMA-1B/3B-Instruct and Qwen2.5-Math-1.5B), as well as the **Random Masking** and **PPL-based Masking** baselines discussed in Section 4.1, are essentially implementations of **Chain-of-Thought (CoT) knowledge distillation** as referenced in the related work.
>
> Specifically, the comparisons you mention—between Random Masking and CoT knowledge distillation, as well as between our proposed CAD method and CoT knowledge distillation—are directly addressed **in Sections 4.1 and 3.2 (Table 1)** respectively. Therefore, we believe our baselines are both strong and appropriate.
>
> We acknowledge that the terminology used to describe “KD” in the manuscript may have caused confusion. To clarify, the baselines employed correspond to CoT knowledge distillation rather than simple answer-based knowledge distillation. We will emphasize this distinction more clearly in future versions of the paper to avoid misunderstandings.
>
> **Q5: Section 2.2 describes token masking and data augmentation with misleading tokens detected individually, but Section 4.2 shows consecutive spans being masked. How are individual misleading tokens combined into spans for masking?**
>
> **A5:** The process of masking misleading tokens consists of two stages.
>
> First, for each instruction, we **compute the normalized gradient difference** between the student and teacher for every token **individually**.
>
> After obtaining these per-token scores, we set a threshold to identify which tokens are considered misleading. If multiple consecutive tokens within the same sample all exceed this threshold, they are grouped and masked together as **a contiguous span, rather than as isolated tokens**.
>
> In this way, while the detection of misleading tokens is performed at the individual token level, the actual masking operation is applied at the span level when adjacent tokens are identified as misleading.
>
> This approach better reflects the reality that misleading patterns can often appear as phrases or segments, preserving the semantic integrity of the data augmentation process.
>
> **Q6: Is random masking in Section 4.1 applied to consecutive token spans or independent tokens, given that misleading token masking targets spans?**
>
> **A6:** In Subsection 4.1, the random‑masking setting mirrors CAD’s span lengths: for any example where CAD masks n consecutive tokens, the baseline also masks n consecutive tokens, with the start position chosen uniformly at random.
>
> **Q7: How were misleading tokens identified in Figure 1’s preliminary experiment? Was the same gradient-based method used as in the main approach?**
>
> **A7:** The misleading tokens in Figure 1 were identified using the gradient‑based attribution technique detailed in the Methods section. This experiment use the same method as the proposed methodology's gradient-based approach.
>
> **Q8: How was the threshold in Equation 6 chosen? The appendix sensitivity analysis is hard to verify; more explanation in the main text is needed.**
>
> **A8:** Refer to **Reviewer 2JsC Q2(a) and Q2(c)**
>
> **Q9: [line 58] typo "Causal Attention Distillation"**
>
> **A9:** Thanks for your suggestion, we will update in the next version.

---

> > ### Comment · Reviewer_WSL6 · 2025-08-04
> >
> > Thank you for your detailed and thoughtful rebuttal. I appreciate the effort made to clarify key aspects of your methodology. Several of my earlier concerns have been addressed, though I believe some critical points still remain.
> >
> > ### Response to A1 and A2
> > I appreciate the clarity in A1 regarding the isolation of CAD's benefits from confounding factors such as sequence length or standard CoT distillation. The ablation studies help clarify CAD’s unique contributions.
> >
> > The explanation in A2 regarding the definition of misleading tokens—based on gradient discrepancies in student-wrong / teacher-correct samples—is useful and contributes to a better understanding of the method. However, since this data selection strategy is central to the approach, I strongly suggest describing it explicitly in the main paper. Additionally, please clarify whether the baselines in Table 1 also use this filtered subset. This distinction affects the fairness of the comparisons.
> >
> > ### Response to Q8
> > Upon reviewing the threshold sensitivity analysis (as described in the responses to reviewer 2JsC), it's clear that CAD's performance can be affected by the choice of τ, with some configurations—e.g., τ=0.05 or 0.15 on MATH—showing performance drops compared to KD w/o Mask. However, this effect appears to be limited in most cases, excluding SQuAD 2.0 where the variance is more pronounced. **It would still be valuable to include a brief explanation in the main text of how τ is selected or tuned, even heuristically, to improve clarity and reproducibility.**
> >
> > ### Response to Causal Framing
> > Given that the method is named "Causal Attention Distillation," it is important that the causal framing plays a meaningful role in the methodology. While the rebuttal attempts to justify the approach through causal language (e.g., treating misleading tokens as confounders and masking as intervention), natural language often involves more complex token-level dependencies, including mediator structures such as `X₁ → A → X₂ → Y`.
> >
> > **In such cases, simply masking a token like A may not break a spurious link, but rather interrupt a valid reasoning path.** For example, if A serves as a condition for X₂ (i.e., the structure is X₁ → A → X₂ → Y), then removing A may sever the necessary dependency between X₁ and X₂, making it impossible to preserve their causal relationship. Therefore, while the proposed causal framing provides an intuitive narrative, **additional quantitative experiments would be needed to support the claim that CAD identifies and intervenes on actual causal relationships within the input.**
> >
> > Overall, the causal interpretation currently enhances the paper at a conceptual level, but its empirical grounding would benefit from further validation. Strengthening this link would increase the rigor and originality of the work.

---

> ### Author Response · Authors · 2025-08-04
> **Response to Reviewer WSL6**
>
> > **Comment (A1):  I appreciate the clarity in A1 regarding the isolation of CAD's benefits from confounding factors such as sequence length or standard CoT distillation. The ablation studies help clarify CAD’s unique contributions.**
>
> **Response 1:** Thank you for your feedback. We're glad the ablation studies in A1 clearly demonstrate CAD’s unique contributions, independent of factors like sequence length and standard CoT distillation. We will retain and highlight these results in the revision.
>
> > **Comment (A2):  The reviewer appreciates the explanation of misleading tokens but requests a clearer description of the data selection process in the main paper and clarification on whether the baselines use the same filtered subset.**
>
> **Response 2:** Thank you for your valuable feedback. We will incorporate a more detailed description of the misleading token selection mechanism into the main paper to improve clarity and transparency, as this component is central to our approach.
>
> Regarding the data used in Table 1, the training configurations are as follows:
>
> - **Baseline (CoT KD):** 12K Original Training Samples
> - **CAD:** 12K Original Training Samples + Counterfactual Samples (generated by pruning misleading spans)
>
> Importantly, the 12K original samples used in **both CoT KD and CAD are unfiltered** and include all cases where the student model succeeds or fails.
>
> This setup ensures **a fair comparison between CAD and the baseline**, as both methods are built upon the same foundational dataset.
>
> > **Comment (Q8):  The reviewer notes that CAD's performance is somewhat sensitive to the choice of the threshold τ, with certain configurations (e.g., τ = 0.05 or 0.15 on MATH) showing performance drops relative to KD w/o Mask. While this effect is limited in most cases, it is more pronounced on SQuAD 2.0. The reviewer suggests including a brief explanation in the main paper of how τ is selected or tuned, even heuristically, to improve clarity and reproducibility.**
>
> **Response 3:** Thank you for the thoughtful suggestion.   We agree that clarifying the selection of the threshold τ would improve the clarity and reproducibility of our method.
>
> Based on our threshold sensitivity analysis on MATH, Code, and SQuAD 2.0, we find that **τ = 0.15 works well across tasks for LLaMA3.2-1B-Instruct, while τ = 0.10 yields better overall results for LLaMA3.2-3B-Instruct**.  We also observe that, regardless of whether the supervision is applied at the instruction or response level, **lower-capacity models are more susceptible to noise from spurious tokens and thus benefit from a stricter threshold**.  This insight guides our heuristic selection of τ in practice.
>
> To improve clarity, we will **add a dedicated subsection in Section 4 of the main paper detailing the selection strategy for τ**, along with a cross-dataset sensitivity analysis that illustrates its impact across different model sizes and tasks.
>
> > **Comment on Causal Framing:  The reviewer raises concerns that natural language involves more complex token-level dependencies, such as mediator structures, where masking a token like A could disrupt valid reasoning paths. Further empirical validation is suggested to support the claim that CAD performs meaningful causal interventions on input tokens.**
>
> **Response 4:** Thank you for the thoughtful and insightful feedback.  We fully agree that token-level dependencies in natural language can involve more complex structures, including mediators such as X₁ → A → X₂ → Y. We address your comment from two perspectives:
>
> First, **regarding the concern that masking a token like A might inadvertently disrupt a valid reasoning path**: in our method, misleading tokens are identified based on cases where the student model initially produces an incorrect answer but becomes correct after the token is masked.  **This selection criterion ensures that the masked token is not part of a valid reasoning chain**—on the contrary, its removal leads to improved model performance.  Thus, the masking operation functions as a targeted intervention that eliminates spurious influences without interfering with necessary dependencies.
>
> Second, we appreciate your observation that our current causal framing may implicitly assume a simplified A → Y structure.  Inspired by your comment, we will revise our description to better reflect more general causal patterns, including mediator chains such as X₁ → A → X₂ → Y. In these cases, **the masked token A may serve as an incorrect condition or irrelevant premise that propagates downstream reasoning errors (e.g., X₂)**.  Removing such tokens **can help disrupt not only direct spurious links  (as in A → Y), but also more complex erroneous intermediate dependencies (e.g., X₁ → A → X₂ → Y)**, thereby aligning with the broader goals of causal intervention.
>
> We will clarify these points in the revised paper to improve the conceptual rigor of our causal framing, while keeping the core method empirically grounded.

---

> ### Author Response · Authors · 2025-08-06
> **Follow-Up: Seeking Further Feedback**
>
> Dear Reviewer WSL6,
>
> We have diligently addressed all the concerns raised during the rebuttal period, and we would like to kindly ask if any remaining issues have been resolved. As the rebuttal deadline is fast approaching, we would greatly appreciate any further feedback, suggestions, or concerns at your earliest convenience. We are eager to engage in further discussion and clarify any unresolved points to ensure all matters are fully addressed.
>
> Best regards,
>
> Authors of Submission 9306

---

> > ### Comment · Reviewer_WSL6 · 2025-08-06
> >
> > Thank you for your thoughtful and detailed responses. I especially appreciate the clarification that CAD primarily targets student-wrong / teacher-correct examples, identifies misleading tokens based on gradient discrepancies, and constructs counterfactual training samples by masking those tokens in span units.
> >
> > I have one additional question regarding the training dataset:
> >
> > As I understand it, CAD generates counterfactual samples by masking spans of misleading tokens from student-wrong / teacher-correct examples, and then performs knowledge distillation using teacher logits on **both** the original 12K training samples and the newly generated counterfactual ones. If the proposed method is intended to function as a causal intervention—removing spurious influences from misleading tokens—wouldn't it be more causally sound to **replace** the original samples that contain misleading tokens, rather than include them alongside the counterfactual ones?
> >
> > In other words, if the presence of misleading tokens genuinely harms the student's reasoning process, then continuing to train on the original samples that include them might undermine the causal objective of CAD. I’d be very interested in hearing the authors’ rationale or experimental justification for choosing to retain the full original dataset in addition to the counterfactual samples.
> >
> > Also, if possible, could you share some statistics on the counterfactual samples—such as the minimum, maximum, and average token length, as well as the total number of counterfactual samples? This would help contextualize the overall influence of these samples within the full training procedure.
> >
> > As the rebuttal period is approaching its end, I will do my best to provide timely feedback in return. Once again, thank you for the serious and constructive discussion.

---

> > > ### Author Response · Authors · 2025-08-06
> > > **Response to Reviewer WSL6**
> > >
> > > >**Q1: What is the rationale or experimental justification for retaining the full original dataset alongside the counterfactual samples, rather than replacing or reducing it?**
> > >
> > > **Response to Q1:**
> > >
> > > Thank you for the insightful question. The rationale for retaining the full original dataset alongside the counterfactual samples lies in the **contrastive learning signal** it provides. By exposing the student model to both the original samples containing misleading patterns and the corresponding counterfactuals with those patterns removed, **the model can learn to downweight its reliance on such patterns during inference and focus instead on more causally relevant information**.
> > >
> > > If the original samples were excluded, the misleading patterns would never appear in the training data, and the model would lack the necessary context to learn not to pay excessive attention to them.  Therefore, keeping the original data—particularly those samples the student initially gets wrong—is essential for enabling the model to distinguish between spurious and meaningful reasoning paths through contrastive supervision.
> > >
> > > On the other hand, we also hope that the student model can still produce correct answers when misleading patterns are present in the data. **We believe achieving this ability requires the model to learn from cases that contain misleading patterns**.
> > >
> > > To further support this rationale, we are conducting an ablation study that removes the original samples during training. As this requires retraining the model, we will provide the results before the rebuttal deadline.
> > >
> > > >**Q2: Request for statistics on counterfactual samples—including the minimum, maximum, and average token length, as well as the total number of samples—to better understand their influence during training**
> > >
> > > **Response to Q2:**
> > >
> > > Thank you for the helpful suggestion.  Below, we report token length statistics and total sample counts for both the original and counterfactual data, across different tasks and model variants.
> > >
> > > **Math Task**
> > >
> > > | Model                          | Min Length | Max Length | Avg Length | Total | AIME | MATH | GSM8K | Olympiad |
> > > |-------------------------------|------------|------------|------------|-------|------|------|--------|-----------|
> > > | Original Samples              | 23         | 2711       | 98.50      | 12000 | 3000 | 3000 | 3000   | 3000      |
> > > | LLaMA3.2-1B-Instruct          | 24         | 2710       | 160.78     | 4576  | 1620 | 1029 | 907    | 1020      |
> > > | LLaMA3.2-3B-Instruct          | 23         | 2711       | 148.79     | 2843  | 1228 | 709  | 172    | 734       |
> > >
> > > **Code Task**
> > >
> > > | Model                          | Min Length | Max Length | Avg Length | Total |
> > > |-------------------------------|------------|------------|------------|--------|
> > > | Original Samples              | 34         | 527        | 140.17     | 12000 |
> > > | LLaMA3.2-1B-Instruct          | 47         | 507        | 158.61     | 6210  |
> > > | LLaMA3.2-3B-Instruct          | 46         | 446        | 159.72     | 4059  |
> > >
> > > We observe that the average length of counterfactual samples is slightly higher than that of the original samples in both Math and Code tasks. This may be due to the fact that longer, more complex problems tend to contain more misleading patterns for LLaMA3.2-1B-Instruct and LLaMA3.2-3B-Instruct, making them more likely to trigger counterfactual generation.
> > >
> > > We will include a summary of these statistics in the revision to improve transparency and help readers better understand the scale and characteristics of the counterfactual data.

---

> > > ### Author Response · Authors · 2025-08-08
> > > **Supplementary Response to Reviewer WSL6**
> > >
> > > >**Q1: What is the rationale or experimental justification for retaining the full original dataset alongside the counterfactual samples, rather than replacing or reducing it?**
> > >
> > >
> > > **Supplementary Response to Q1:**
> > >
> > > To directly evaluate the necessity of retaining the full original dataset alongside the counterfactual samples, we conducted an ablation study with the following configurations:
> > >
> > > * **CoT KD:** 12K original training samples
> > > * **Ablation:** Original samples where the student model predicts correctly + counterfactual samples
> > > * **CAD:** 12K original training samples + counterfactual samples (generated by pruning misleading spans)
> > >
> > > **Results:**
> > >
> > > | Model             |   GSM8K   |    Math   |  Olympiad |    Avg.   |
> > > | ----------------- | :-------: | :-------: | :-------: | :-------: |
> > > | LLaMA-1B CoT KD   |   56.79   |   33.40   |    8.90   |   33.03   |
> > > | LLaMA-1B Ablation | **58.15** |   34.80   |    7.42   |   33.46   |
> > > | LLaMA-1B CAD      |   57.70   | **35.40** | **10.09** | **34.40** |
> > > | LLaMA-3B CoT KD   |   82.87   |   49.00   |   18.99   |   50.29   |
> > > | LLaMA-3B Ablation | **84.08** |   47.80   |   16.02   |   49.30   |
> > > | LLaMA-3B CAD      |   83.09   | **51.80** | **20.77** | **51.88** |
> > >
> > > **Key observations:**
> > >
> > > 1. **CAD outperforms Ablation overall**, demonstrating that retaining original samples containing misleading patterns is beneficial.
> > > 2. The slight advantage of Ablation on GSM8K is due to dataset composition:
> > >
> > >    * GSM8K has **fewer “student-wrong” cases** than MATH or Olympiad, so their removal has a smaller negative effect.
> > >    * GSM8K is simpler than MATH and Olympiad. In Ablation, the “student-correct” subset contains more GSM8K samples, increasing its relative proportion compared to CAD. **This reduces the share of harder datasets and biases optimization toward easier tasks**.
> > >
> > > **Data composition:**
> > >
> > > | Setting      | Model    | GSM8K | MATH | Olympiad | AIME |
> > > | ------------ | -------- | ----- | ---- | -------- | ---- |
> > > | **Ablation** | LLaMA-1B | 2801  | 2252 | 1439     | 2285 |
> > > |              | LLaMA-3B | 2927  | 2723 | 1582     | 2556 |
> > > | **CAD**      | LLaMA-1B | 3907  | 4029 | 4020     | 4620 |
> > > |              | LLaMA-3B | 3172  | 3709 | 3734     | 4228 |
> > >
> > > These findings support our **twofold rationale**:
> > >
> > > (1) Retaining the full original dataset alongside the counterfactual samples provides a contrastive learning signal, enabling the student model to downweight reliance on misleading patterns and focus on causally relevant information;
> > >
> > > (2) Retaining such data enables the model to produce correct answers even when misleading patterns are present, which requires exposure to cases that contain these patterns.
> > >
> > >
> > >
> > > ---
> > >
> > >
> > >
> > > We have diligently addressed all points raised during the rebuttal period, and with the deadline fast approaching, we would greatly appreciate any remaining feedback or suggestions at your earliest convenience.
> > >
> > > If our responses have fully resolved the issues, we would be grateful if this could be reflected in your final justification.
> > >
> > > Best regards,
> > >
> > > Authors of Submission 9306

---

> ### Comment · Reviewer_WSL6 · 2025-08-08
>
> I thank the authors for their detailed and constructive engagement throughout the rebuttal period. Many of my earlier concerns have been addressed, and I appreciate the additional statistics and clarifications.
>
> Regarding **Q2**, I agree with the authors’ observation that counterfactual samples tend to be longer; this is an interesting property of the data, and I find their explanation plausible.
>
> However, for **Q1** and the associated experiments, I still have several reservations:
> 1. On the claim of a contrastive learning effect from using both original and counterfactual data
> I find it difficult to agree with this claim. Since both the original and counterfactual data are trained with a CE loss to maximize log-likelihood, the original data will still encourage the model to generate misleading tokens. **The mere use of original and counterfactual data together does not necessarily create a contrastive signal** in the standard CE setup, and this connection remains unclear to me.
> 2. On the claim that the model can learn to ignore misleading patterns when both datasets are used
> I partially agree. The results suggest that retaining misleading patterns can outperform fully removing them, but this is somewhat counterintuitive—**if misleading tokens induce spurious correlations, why should preserving them be beneficial?** This point may be outside the immediate scope of the paper, but it warrants further discussion or at least acknowledgement.
>
> From a broader perspective, I personally find the method closer to reasoning distillation with pruning than to a strict causal framework, though the authors’ interpretation is not without merit. The planned revision to better align the causal framing with complex token-level dependencies is welcome.
>
> I also encourage the authors to revise lines 125–126 to explicitly state the composition and intended role of the training data, as this will help readers fully understand the design choices. Including results from the ongoing ablation study that removes original samples would also be valuable if available before the camera-ready deadline.
>
> Overall, I appreciate the diverse experiments and analyses, which have addressed most of my concerns. I will raise my score accordingly. While I personally view the approach as closer to **reasoning distillation with pruning** than to a pure causal framework, the authors’ claim is also persuasive. I hope the discussed data statistics, the clarification of the training data composition and intention (particularly lines 125–126), and the additional ablation results will be well reflected in the revision.

---

> > ### Author Response · Authors · 2025-08-09
> > **Follow-up on Remaining Concerns – Submission 9306**
> >
> > **Dear Reviewer WSL6**,
> >
> > Thank you again for your valuable feedback and for the positive update to the score.
> >
> > We also appreciate the additional concerns you raised earlier, and we have provided detailed responses addressing them in our follow-up.
> >
> > As the discussion period now has **less than 10 hours** remaining, we wanted to kindly check whether our clarifications have addressed the concerns you mentioned in your last discussion post, or if there is anything else we can further elaborate on before the deadline.
> >
> > If our responses have resolved your remaining concerns, we would be grateful if this could be taken into consideration in your final justification.
> >
> > Best regards,
> >
> > Authors of Submission 9306

---

> ### Author Response · Authors · 2025-08-08
> **Further Clarification on Q1 Reservations**
>
> **Further Clarification on Q1**
>
> We appreciate your thoughtful comments and would like to clarify **two key points regarding potential misunderstandings about misleading patterns**.
>
> **First, in our setting—whether at the instruction or response level—misleading patterns exist only in the input, not in the output.**
>
> Therefore, the concern that “the original data will still encourage the model to generate misleading tokens” does not apply, as no misleading tokens appear in the ground-truth outputs.
>
> Likewise, the question “if misleading tokens induce spurious correlations, why should preserving them be beneficial?” assumes that CAD preserves spurious correlations. In fact, CAD is designed to **reduce** them.
>
> By jointly training on original samples and their counterfactual counterparts, the student model learns to downweight attention to misleading patterns—a conclusion supported not only by CAD’s consistently higher performance than CoT KD, but also by the qualitative evidence in Section 4.3 (Case Study). Without this dual exposure, spurious correlations in the student model would remain unmitigated.
>
> **Second, misleading patterns do not cause spurious correlations in the teacher model’s reasoning.**
>
> These patterns are features that the student over-attends to, but that the teacher ignores, and the teacher predicts correctly regardless of their presence (by definition, all original samples are correctly solved by the teacher). Since the student is distilled from the teacher via logits alignment, the training signal itself does not introduce spurious correlations.
>
> Finally, while the CE loss itself is unchanged, **the contrastive learning effect arises from the paired input construction with identical labels**: each original–counterfactual pair presents the same task—with and without misleading patterns—forcing the student to adjust its attention distribution. Attending to misleading patterns harms performance on counterfactuals, while ignoring them benefits both, creating a functional contrastive signal in the input space.
>
> In short, misleading patterns do not cause spurious correlations in the teacher model’s reasoning.  Instead, the dual exposure to original and counterfactual inputs enables the student to recognize and suppress such correlations in its own reasoning process.
>
> -------------------------
> We sincerely thank you for the positive feedback and for raising the score.
>
> We also appreciate your recognition of our efforts in conducting diverse experiments and analyses, as well as your constructive suggestions.
>
> We will ensure that the clarification of the training data composition and intention (particularly lines 125–126), the discussed data statistics, and the additional ablation results are clearly and thoroughly incorporated in the revised version of the paper.
>
> Best regards,
>
> Authors of Submission 9306

---

### Official Review · Reviewer_ZMF9 · 2025-07-03

**Clarity:** 3
**Significance:** 2
**Originality:** 2
**Rating:** 4
**Confidence:** 3

**Summary:**

This paper introduces Causal Attention Distillation (CAD), a novel two-stage distillation framework that aims to improve reasoning consistency in large language models (LLMs). The authors adopt a causal perspective and treat misleading tokens as spurious confounders. CAD first identifies such misleading tokens through gradient-based sensitivity comparisons between a teacher and a student model. Then, these tokens are masked during training, enabling the student model to better align its attention with the teacher's on causally relevant parts of the input. The method is evaluated on both mathematical reasoning and code generation task, showing consistent performance gains over standard knowledge distillation and other masking strategies (random, PPL-based).

**Questions:**

1.How to ensure the stability of the gradient difference threshold (τ_mis) in cross-task scenarios?

2.How does CAD compare to standard KD in terms of training time and memory usage? Are the additional costs of gradient comparison practical for large-scale applications? Could the gradient comparison be approximated more efficiently?

**Ethical Concerns:**

["NO or VERY MINOR ethics concerns only"]

**Final Justification:**

After reading the authors' response and other reviewers' comments, my concerns have been addressed, thus I change my score from 3 to 4.

**Limitations:**

yes

**Quality:**

2

**Strengths And Weaknesses:**

Strengths: 1.The author combines causal inference with gradient analysis to effectively separate false correlations, providing a new approach for addressing attention bias in LLM reasoning.

2.The paper systematically compares different masking strategies (random, PPL-based, gradient-based), as well as varying the number of response segments. These analyses strengthen the empirical claims and justify the design choices.

3.The attention heatmap comparison between CAD and traditional KD adds interpretability, demonstrating how CAD reshapes the student’s focus toward causally relevant information，the case study (Section 4.3) clearly demonstrates the attention shift toward critical tokens after CAD training, helping to validate the underlying causal intuition.

Weaknesses: 1.The method was mainly verified on the LLaMA and Qwen series models, and the generalization of other architectures (such as GPT, Claude) was not tested.

2.The mathematical proof of the causal model is relatively brief and does not fully expound the theoretical connection between the DAG structure and attention optimization.

3.The proposed method relies on computing token-level gradients for both teacher and student, which may incur high computational overhead. No time/memory cost analysis is provided.

4.The decision to mask individual misleading patterns instead of all at once is motivated by empirical performance. However, the semantic trade-off (between removing noise and retaining context) could be formalized further, perhaps via information-theoretic analysis.

---

> ### Author Rebuttal · Authors · 2025-07-31
>
> **Q1: The method was mainly verified on the LLaMA and Qwen series models, and the generalization of other architectures (such as GPT, Claude) was not tested.**
>
> **A1:** We acknowledge that our method was primarily validated on the LLaMA and Qwen model families.
>
> **Our approach aligns with the prevailing trend in recent model capability enhancement research, which mainly focuses on open-source models**. Closed-source models like GPT and Claude, however, restrict access to gradient information, which is crucial for identifying misleading tokens in our method. This practical constraint prevents direct application of our approach to these architectures.
>
> Furthermore, using teacher and student models from the same family (e.g., LLaMA-3.2-1B-Instruct and LLaMA-3.3-70B-Instruct) provides significant advantages. **Their shared architecture and similar pretraining data create compatible internal representations, enabling the student model to more effectively learn from the teacher through distillation**. This enhances the student’s ability to focus attention on critical information and ultimately improves reasoning performance.
>
> **Q2: The mathematical proof of the causal model is relatively brief and does not fully expound the theoretical connection between the DAG structure and attention optimization.**
>
> **A2:** Thank you for this suggestion. To make the link between our DAG formulation and attention optimization clearer, we will expand the theoretical derivation as follows:
>
> Our goal is to **eliminate spurious correlations** introduced by confounding tokens $\(A\)$. In the presence of a confounder $\(A\)$ that influences both $\(X_i\)$ and the output $\(Y\)$ (dashed arrows in Figure 3), the standard conditional distribution becomes:
>
>
> $P(Y | X_i = x) = \sum_{a} P(Y \mid X_i = x, A = a) \, P(A = a \mid X_i = x)$
>
>
> which departs from the true interventional distribution $\(P(Y \mid \mathrm{do}(X_i = x))\)$. The discrepancy arises because the term \$(P(A \mid X_i)\)$ carries spurious influence from the path $\(A \to Y\)$.
>
> We perform **causal pruning** by intervening with $\(\mathrm{do}(A=\emptyset)\)$—i.e., masking out tokens in \(A\). In Pearl’s framework, this blocks the non-causal paths through \(A\), yielding:
>
>
> $P(Y \mid \mathrm{do}(X_i = x)) = P(Y \mid X_i = x, A=\emptyset)$,
>
>
> which restores the genuine causal effect of $\(X_i\)$ on $\(Y\)$.
>
> In transformer models, **attention weights** serve as a soft selection mechanism over input tokens: higher attention on \(X_i\) implies a stronger dependency of the output on that token. By training on these interventional (counterfactual) inputs, the student’s attention distribution is **optimized** to focus on true causal parents—and away from confounders—thereby aligning its internal reasoning with the DAG’s structure.
>
> This causal pruning both improves robustness by removing misleading cues and enhances interpretability: as demonstrated in our case study, CAD’s gradient distributions focus more sharply on the instruction’s key tokens compared to standard KD, providing clearer, more explainable reasoning behavior.
>
>
> **Q3: The proposed method relies on computing token-level gradients for both teacher and student, which may incur high computational overhead. No time/memory cost analysis is provided. How does CAD compare to standard KD in terms of training time and memory usage? Are the additional costs of gradient comparison practical for large-scale applications? Could the gradient comparison be approximated more efficiently?**
>
> We quantified the additional computational overhead of CAD compared to standard KD by measuring end-to-end runtime using identical hardware:
>
> **Gradient computation (one-time, 8× NVIDIA A100 80GB):**
> Computing gradients for 7K samples consistently takes about 3 hours, regardless of the student model size (LLaMA‑3.2‑1B, LLaMA‑3.2‑3B, or Qwen2.5‑Math‑1.5B), when using larger teacher models (LLaMA‑3.3‑70B or Qwen2.5‑72B-Instruct).
>
> **Counterfactual generation (one-time, offline via vLLM, 4×NVIDIA A100 80GB):**
> Generating counterfactuals for 26K samples requires around 50 minutes for smaller models (e.g., LLaMA‑1B) and approximately 2.85 hours for larger models (e.g., LLaMA‑70B or Qwen2.5‑72B). This process is easily parallelizable.
>
> **Training overhead (3 epochs 4×NVIDIA A100 80GB):**
> | Model                  | KD Runtime | CAD Runtime | Overhead  |
> |------------------------|-----------:|------------:|----------:|
> | LLaMA‑3.2‑1B           |     26.2 h |      29.2 h |   +11.5%  |
> | LLaMA‑3.2‑3B           |     23.7 h |      26.2 h |   +10.6%  |
> | Qwen2.5‑Math‑1.5B      |     27.3 h |      30.9 h |   +13.3%  |
>
> Overall, CAD incurs only 10–13% additional training overhead compared to standard KD, while achieving 2–3% absolute accuracy improvements in mathematics and coding tasks. Importantly, the extra data-processing steps—gradient normalization, span pruning, and counterfactual generation—are one-time, offline operations, fully parallelizable, and do not recur during inference or downstream fine-tuning, making CAD practical to scale.
>
> **Q4:The decision to mask individual misleading patterns instead of all at once is motivated by empirical performance. However, the semantic trade-off (between removing noise and retaining context) could be formalized further, perhaps via information-theoretic analysis.**
>
> **A4:** We appreciate the suggestion and agree that the trade-off between removing noise and preserving semantic context is an important aspect of our method.
>
> Our choice to mask misleading patterns incrementally—rather than masking all identified tokens at once—was guided by empirical observations: masking too aggressively, especially in instructions with limited length, often harms performance by disrupting essential context or breaking syntactic structures. In contrast, span masking allows the model to selectively suppress misleading cues while retaining enough context to preserve semantic coherence.
>
> We fully agree that this semantic trade-off could benefit from formalization through information-theoretic approaches. For example, one could quantify the information gain associated with varying masking proportions to identify an optimal point that balances noise suppression and context retention. Additionally, metrics such as predictive entropy or the KL divergence between teacher and student attention distributions could serve as principled indicators for optimal masking strategies.
>
> Although our current strategy relies primarily on empirical evidence, we view a formal, information-theoretic analysis of this trade-off as an important and promising direction for future research. We greatly appreciate the reviewer highlighting this valuable perspective.
>
> **Q5: How to ensure the stability of the gradient difference threshold (τ_mis) in cross-task scenarios?**
>
> **A5:** To evaluate the stability of the gradient difference threshold(τ_mis) across different tasks, we measured each model’s accuracy on three benchmarks—Math, Code, and SQuAD 2.0—at (τ_mis)values of 0.05, 0.10, and 0.15:
>
> | Task        | Model               | τ=0.05  | τ=0.10  | τ=0.15  |
> |-------------|---------------------|:-------:|:-------:|:-------:|
> | **Math**    | LLaMa‑1B‑Instruct   | 32.43%  | **34.40%**  | 33.17%  |
> |             | LLaMa‑3B‑Instruct   | **51.88%**  | 51.07% | 50.55%  |
> | **Code**    | LLaMa‑1B‑Instruct   | 17.85%  | 18.27%  | **19.76%**  |
> |             | LLaMa‑3B‑Instruct   | 32.83%  | **33.55%**  | 32.95%  |
> | **SQuAD 2.0** | LLaMa‑1B‑Instruct  | 72.12%  | 73.33%  | **74.78%**  |
> |             | LLaMa‑3B‑Instruct   | 88.67%  | **89.44%**  | 83.66%  |
>
> We observe that LLaMa‑1B‑Instruct generally achieves its best performance at \( τ= 0.15\), while LLaMa‑3B‑Instruct consistently performs best at \(τ = 0.10\) across these tasks. This indicates that the threshold \(τ_mis\) is relatively stable in cross-task scenarios.

---

> ### Author Response · Authors · 2025-08-05
> **Seeking Further Discussion on Rebuttal**
>
> Dear Reviewer ZMF9:
>
> Thank you again for the great efforts and valuable comments. We have carefully addressed the main concerns in detail. We hope you might find the response satisfactory (similar as the other reviewers). As the discussion phase is about to close, we are very much looking forward to hearing from you about any further feedback. We will be very happy to clarify any further concerns (if any).

---

### Official Review · Reviewer_qdQ9 · 2025-07-04

**Clarity:** 4
**Significance:** 3
**Originality:** 3
**Rating:** 4
**Confidence:** 4

**Summary:**

This paper proposes Causal Attention Distillation (CAD), a two-stage framework for enhancing the reasoning ability and interpretability of student large language models (LLMs) by distilling causal attention from a more advanced teacher. In the first stage, distracting or confounding tokens in training data are identified through gradient-based comparisons between teacher and student models. In the second stage, the student model is distilled not only on original inputs but also on counterfactual samples created by pruning confounders, using a hybrid loss. Experimental results on mathematical reasoning and code benchmarks indicate that CAD improves student model accuracy and shifts attention toward more causally relevant tokens, enhancing robustness and interpretability.

**Questions:**

- How sensitive is confounding token identification in CAD to potential errors or misattributions in the teacher model? If the teacher misfocuses or mispredicts, could CAD propagate harmful attention patterns?
- The method depends on clear differences between teacher and student gradient attributions. Have the authors tested robustness under adversarially inserted confounders or in noisy real text, especially for models not tuned for math/code?

**Ethical Concerns:**

["NO or VERY MINOR ethics concerns only"]

**Final Justification:**

The initial score still reflects my evaluation to this paper after carefully checking out the authors' rebuttal.

**Limitations:**

Yes

**Quality:**

3

**Strengths And Weaknesses:**

**Strengths:**

- The paper is motivated pretty well and carefully written.
- Experiments span multiple settings and benchmarks. The main results show consistent accuracy gains over standard knowledge distillation (KD), especially when both instruction and response-level masking are used.


**Weaknesses:**

- CAD’s reliance on a large and advanced teacher to guide the token importance. This limits its application when such teacher models are unavailable or computationally prohibitive.
- The proposed approach requires teacher-student training and extra steps to compute and normalize gradients, generate counterfactuals, and perform span pruning. This could add extra computational overhead that can't be omitted when scaling up?
- It seems the magnitude of improvement over standard KD (about 2-3% absolute) is rather moderate. For the largest/strongest teacher-student pairs, the relative improvement is smaller.
- The efficacy of the proposed method is assessed mainly on mathematical and code-generation tasks where the reasoning is relatively structured. I wonder if other scenarios like open-domain QA and long-text summarization (I later noticed the authors mentioned this as well in the limitation section).

---

> ### Author Rebuttal · Authors · 2025-07-31
>
> **Q1: CAD’s reliance on a large and advanced teacher to guide the token importance. This limits its application when such teacher models are unavailable or computationally prohibitive.**
>
> **A1:** We acknowledge in **Appendix A.8 Limitation** that CAD’s reliance on a large, advanced teacher model is indeed a limitation. However, this dependency does not diminish the value of our approach. Beyond the well-known gap in reasoning capabilities, smaller models also lack a critical ability to focus attention on key information. CAD enhances standard distillation by guiding small models to focus on the critical information large models use during reasoning. Such guidance is especially crucial for enabling smaller models to effectively handle challenging reasoning tasks.
>
> Furthermore, **our experiments in Section 4.2 validate the importance of teacher models**. Specifically, CAD is compared against random masking and PPL-masking on Olympiad problems. The results show that PPL-masking performs similarly to or worse than random masking and standard distillation, **indicating that smaller models without external guidance often fail to identify genuinely important tokens in complex reasoning tasks**. By leveraging teacher models to explicitly highlight these essential tokens, CAD substantially improves the reasoning performance of smaller models, thereby unlocking their greater potential.
>
> **Q2: The proposed approach requires teacher-student training and extra steps to compute and normalize gradients, generate counterfactuals, and perform span pruning. This could add extra computational overhead that can't be omitted when scaling up?**
>
> **A2:** We quantified the additional computational overhead of CAD compared to standard KD by measuring end-to-end runtime using identical hardware:
>
> **Gradient computation (one-time, 8× NVIDIA A100 80GB):**
> Computing gradients for 7K samples consistently takes about 3 hours, regardless of the student model size (LLaMA‑3.2‑1B, LLaMA‑3.2‑3B, or Qwen2.5‑Math‑1.5B), when using larger teacher models (LLaMA‑3.3‑70B or Qwen2.5‑72B-Instruct).
>
> **Counterfactual generation (one-time, offline via vLLM, 4×NVIDIA A100 80GB):**
> Generating counterfactuals for 26K samples requires around 50 minutes for smaller models (e.g., LLaMA‑1B) and approximately 2.85 hours for larger models (e.g., LLaMA‑70B or Qwen2.5‑72B). This process is easily parallelizable.
>
> **Training overhead (3 epochs 4×NVIDIA A100 80GB):**
> | Model                  | KD Runtime | CAD Runtime | Overhead  |
> |------------------------|-----------:|------------:|----------:|
> | LLaMA‑3.2‑1B           |     26.2 h |      29.2 h |   +11.5%  |
> | LLaMA‑3.2‑3B           |     23.7 h |      26.2 h |   +10.6%  |
> | Qwen2.5‑Math‑1.5B      |     27.3 h |      30.9 h |   +13.3%  |
>
> Overall, CAD incurs only **10–13%** additional training overhead compared to standard KD, while achieving 2–3% absolute accuracy improvements in mathematics and coding tasks. Importantly, the extra data-processing steps—gradient normalization, span pruning, and counterfactual generation—are one-time, offline operations, fully parallelizable, and do not recur during inference or downstream fine-tuning, making CAD practical to scale.
>
> **Q3: It seems the magnitude of improvement over standard KD (about 2-3% absolute) is rather moderate. For the largest/strongest teacher-student pairs, the relative improvement is smaller.**
>
> **A3:** We believe that the observed 2–3% absolute improvement over standard KD is substantial given the strong performance of **the CoT KD baselines**.
> Specifically, **for the Code task**, standard KD itself achieves an average improvement of **3.86%** over the base models, while our CAD approach further adds an average improvement of **2.39%** on top of KD—indicating that CAD provides gains nearly on par with those of KD itself.
>
> **For Math tasks**, COT KD is already a strong baseline, yet CAD consistently yields an additional 2–3% absolute improvement, clearly demonstrating meaningful incremental value.
>
> Moreover, as shown in our supplementary results on **SQuAD v2** (see Q4), **CAD achieves an average 4% improvement** on QA tasks—highlighting its **robustness and general applicability** across Math, Code, and QA domains.
>
> **Q4: The efficacy of the proposed method is assessed mainly on mathematical and code-generation tasks where the reasoning is relatively structured. I wonder if other scenarios like open-domain QA and long-text summarization (I later noticed the authors mentioned this as well in the limitation section).**
>
> **A4:** We acknowledge in Appendix A.8 that, due to small models’ current limitations with long texts, CAD has constraints in the long‑document domain.
> To test CAD beyond math and code, we evaluated on the Stanford Question Answering Dataset 2.0 (SQuAD 2.0), where each answer is a text span from the passage (or the question may be unanswerable). We sampled 3.5 K training examples and measured exact‑match (EM) on 900 test items.
>
> **Results (EM):**
>
> **Llama‑1b‑Instruct:** 41.11% → KD: 72.56% → CAD: 74.78%
>
> **Llama‑3b‑Instruct:** 73.67% → KD: 83.67% → CAD: 89.44%
>
> These results show that, with sufficient training data, CAD not only boosts performance on math and code tasks but also generalizes effectively to open‑domain QA.
>
> **Q5: The method depends on clear differences between teacher and student gradient attributions. Have the authors tested robustness under adversarially inserted confounders or in noisy real text, especially for models not tuned for math/code?**
>
> **A5:** To evaluate the robustness of our method, we conducted analyses from both the training and testing perspectives.
>
> First, regarding **training data**, we used standard and widely adopted datasets—NuminaMath for math, AceCoder for code, and SQuAD 2.0 for question answering. Our approach constructs distillation data from these clean corpora and **does not introduce noise during training**. Moreover, due to the **inherent capability gap between teacher and student models**, clear differences in gradient attributions naturally arise on these standard datasets, ensuring the generality of our method without requiring adversarial noise.
>
> Second, to assess **test-time robustness**, we introduced noise into evaluation datasets—including Math benchmarks (GSM8K, MATH-500, OlympiadBench) and SQuAD 2.0 (900 samples)—via back-translation, a common data augmentation technique for generating realistic input perturbations. The following table summarizes the results, demonstrating that our method maintains strong performance improvements even under noisy test conditions:
>
> | Dataset             | Model                  | Instruct | KD     | CAD(Instr Mask)    | CAD (Instr & Resp Mask) |
> |---------------------|------------------------|----------|--------|--------|------------|
> | **SQuAD 2.0**       | LLaMA‑3.2‑1B‑Instruct  | 45.89%   | 66.33% | 69.11% | –          |
> |                     | LLaMA‑3.2‑3B‑Instruct  | 69.22%   | 77.00% | 79.44% | –          |
> | **MathBench GSM8K** | LLaMA‑3.2‑1B‑Instruct  | 39.65%   | 50.42% | 51.10% | 51.71%     |
> |                     | LLaMA‑3.2‑3B‑Instruct  | 70.43%   | 74.00% | 74.07% | 76.12%     |
> | **MathBench MATH-500** | LLaMA‑3.2‑1B‑Instruct | 24.80%  | 31.80% | 32.00% | 34.20%     |
> |                     | LLaMA‑3.2‑3B‑Instruct  | 40.80%   | 48.20% | 50.20% | 51.20%     |
> | **MathBench Olympiad** | LLaMA‑3.2‑1B‑Instruct | 3.86%   | 5.19%  | 6.53%  | 6.97%      |
> |                     | LLaMA‑3.2‑3B‑Instruct  | 9.64%    | 14.09% | 15.58% | 19.88%     |
>
> These results demonstrate that CAD remains robust even when noisy inputs are introduced at test time, maintaining and often improving upon standard KD performance. We thank the reviewer for highlighting this important aspect of robustness.
>
>
> **Q6: How sensitive is confounding token identification in CAD to potential errors or misattributions in the teacher model? If the teacher misfocuses or mispredicts, could CAD propagate harmful attention patterns?**
>
> **A6:** Our method includes** a filtering step** to ensure the reliability of misleading token identification. Specifically, we only retain cases **where the teacher model predicts the correct answer both before and after the identified tokens are removed**. This ensures that the removed tokens are not essential for correct reasoning, and that the resulting patterns do not reflect harmful or erroneous attention.

---

> > ### Comment · Reviewer_qdQ9 · 2025-08-08
> >
> > I would like to thank the authors for the additional results and discussions. After careful reconsideration, I think my original score still reflect my evaluation for this paper, so I am keeping it unchanged.

---

> > > ### Author Response · Authors · 2025-08-09
> > > **Follow-up on Rebuttal Feedback – Submission 9306**
> > >
> > > **Dear Reviewer qdQ9,**
> > >
> > > Thank you for your thoughtful review and the constructive concerns you raised.
> > > During the rebuttal, we made substantial efforts to address them, particularly by:
> > >
> > > * **Validating CAD’s effectiveness in the QA domain** (A4) with additional experiments.
> > > * **Conducting extensive robustness evaluations** of CAD (A5) across diverse settings.
> > > * **Providing a detailed time-efficiency comparison** of CAD versus baselines (A2).
> > >
> > > We believe these additions not only directly respond to your comments but also strengthen the paper’s empirical depth and practical relevance. While we understand that you are keeping your original score, we hope the improvements and clarifications we provided will be reflected in your overall impression of the work.
> > >
> > > Thank you again for your time and constructive input—it has been valuable in helping us improve the quality and clarity of our paper.
> > >
> > > Best regards,
> > >
> > > Authors of Submission 9306

---

> ### Author Response · Authors · 2025-08-05
> **Seeking Further Discussion on Rebuttal**
>
> Thank you again for the great efforts and valuable comments. We have carefully addressed the main concerns in detail. We hope you might find the response satisfactory. As the discussion phase is about to close, we are very much looking forward to hearing from you about any further feedback. We will be very happy to clarify any further concerns (if any).

---

> ### Comment · Area_Chair_co2r · 2025-08-06
> **Official Comment from AC**
>
> Dear Reviewer,
>
> As the Area Chair handling this submission, I would like to express my appreciation for the thorough and insightful comments you have provided. The authors have now submitted a comprehensive rebuttal addressing each of the points raised in your reviews.
> To facilitate a fair and rigorous decision-making process, I kindly urge each reviewer to carefully assess whether the authors' rebuttal adequately addresses your specific concerns and technical questions. Your active participation and constructive input during the upcoming discussion phase will be crucial in arriving at a well-informed decision.
>
> Please feel free to share any additional thoughts or clarifications you may have as we proceed with our deliberations.
>
> Best regards,
>
> Your AC

---

> ### Author Response · Authors · 2025-08-06
> **Looking for Further Discussion and Feedback**
>
> Dear Reviewer qdQ9,
>
>
> We have diligently addressed all concerns raised during the rebuttal period, particularly by **validating the effectiveness of the CAD method in the QA domain (A4)**, **conducting extensive robustness evaluations of CAD (A5)**, and **providing a detailed time-efficiency comparison of CAD versus baseline approaches (A2)**.  As it has now been **nearly six days** since our rebuttal, we kindly ask you to review the updated manuscript and let us know if any issues remain.  We greatly appreciate any further comments or suggestions at your earliest convenience and are happy to clarify any outstanding points.
>
>
> Best regards,
>
> The authors of Submission 9306

---

### Official Review · Reviewer_2JsC · 2025-07-07

**Clarity:** 3
**Significance:** 2
**Originality:** 3
**Rating:** 4
**Confidence:** 3

**Summary:**

This paper describes Causal Attention Distillation (CAD), a novel two-stage framework that enhances reasoning in Large Language Models (LLMs) by eliminating attention to spurious markers. Through gradient-guided comparisons between teacher and student models (e.g., LLaMA3.3-70B → LLaMA3.2-1B), CAD first identifies misleading tokens that interfere with causal reasoning, and then masks them during distillation to enhance attention to critical contexts. Based on Perl's causal model, CAD achieves d-separation of confounding variables for consistent gains. This work combines causal theory with practical distillation techniques to provide a scalable solution for achieving robust attention adjustment in LLM.

**Questions:**

Q1. The paper states that spurious confounders are a general issue for LLMs, but Figures 1 and 2 only demonstrate this problem in a 1B model—not in the 70B variant. Could the authors clarify whether this issue is specific to low-parameter models or distillation settings, rather than a universal phenomenon across model scales?

Q2. The paper lacks analysis of the threshold parameter τ, which plays a key role in identifying misleading tokens. Could the authors provide insights into how τ was selected, whether negative values were considered, and how sensitive the model’s performance is to different τ values across datasets?

Q3. Sections 4.1 and 4.2 present experimental results, but it is unclear which model(s) these results are based on. Could the authors specify the model architecture and size used in each subsection to improve reproducibility?

Q4. In Table 1, the proposed methods fall behind some other methods in some datasets—for example, the Instruct Model or KD w/o Mask occasionally outperforms the masking-based methods in LeetCode and Livecode-Bench. Could the authors analyze or explain the possible reasons behind these anomalies?

**Ethical Concerns:**

["NO or VERY MINOR ethics concerns only"]

**Limitations:**

Yes

**Quality:**

3

**Strengths And Weaknesses:**

Strengths

- The paper effectively identifies and addresses LLMs’ susceptibility to spurious tokens during reasoning by introducing the Causal Attention Distillation (CAD) framework. This method improves attention robustness and aligns well with causal principles.

- The paper is clearly structured, with causal graphs, heatmaps, and a well-designed case study that highlights how CAD focuses on critical information and avoids misleading tokens. These visual and narrative aids make the method accessible.

- CAD achieves consistent improvements across models and tasks, showing strong practical utility. Its compatibility with existing distillation workflows allows for easy integration and enhances interpretability.

Weaknesses

-  While the paper claims spurious confounders are common across LLMs, Figures 1 and 2 only show this issue in a 1B model—not in the 70B model. This suggests the problem may stem from insufficient model capacity rather than a universal issue, and should be framed within low-parameter or distillation contexts.
- The paper lacks parameter analysis, especially regarding the threshold τ. τ critically affects performance and generalization; intuitively, it should be negative, as τ = 0 implies equal token sensitivity between student and teacher, contradicting the masking purpose. Its optimal value may also vary across datasets.
- Some experimental details are unclear, such as which model is used in Sections 4.1 and 4.2. This ambiguity affects the reproducibility and clarity of the experimental setup.

---

> ### Author Rebuttal · Authors · 2025-07-31
>
> **Q1: The paper states that spurious confounders are a general issue for LLMs, but Figures 1 and 2 only demonstrate this problem in a 1B model—not in the 70B variant. Could the authors clarify whether this issue is specific to low-parameter models or distillation settings, rather than a universal phenomenon across model scales?**
>
> **A1:** We would like to clarify that spurious confounders are not specific to low‑parameter models or distillation settings, but are a universal phenomenon across model scales. To demonstrate this, we extend the analytical experiments from the Introduction (Appendix A.2.2) to larger models, such as **Llama‑3.3‑70B‑Instruct** and **Qwen2.5‑72B‑Instruct**, using the same 12 K training examples. This enables the models to identify misleading tokens without relying on a teacher model. The specific experimental procedure is as follows:
> First, we select the subset where the large model produces incorrect results. Then, we compute each sample’s gradient of the model to identify potential misleading patterns. Finally, we remove these misleading patterns from the instruction and re‑evaluate the model’s prediction.
>
> | Model                         | Threshold | Originally Wrong | Corrected After Removing Confounders | Improvement |
> |-------------------------------|-----------|-----------------:|-------------------------------------:|------------:|
> | Llama‑3.3‑70B‑Instruct        | 0.10      |            2,112 |                                  453 |      21.45% |
> | Qwen2.5‑72B‑Instruct          | 0.10      |            2,604 |                                  613 |      23.54% |
>
>
> The observed **21.45%~23.54%** improvement on 70 B models after confounder removal **demonstrates that spurious confounders also degrade the reasoning accuracy of large-scale models**. Therefore confirming that spurious confounders are a universal LLM phenomenon, not an artifact of small‑scale or distillation‑only settings.
>
> **Q2(a)：How was the threshold parameter τ selected?**
>
> **A2(a)**: Due to space limitations in the main text, a detailed analysis of the threshold parameter τ is provided in the supplementary material, specifically in Appendix A.1, titled "Threshold Sensitivity Analysis."
>
> In this analysis, we considered both instruction-level and response-level performance. Our findings are as follows:
>
> **1.Instruction-level**: LLaMa3.2-CAD-1B performs best at a threshold of 0.10, while LLaMa3.2-CAD-3B performs best at 0.05.
>
> **2.Response-level**: LLaMa3.2-CAD-1B performs best at 0.15, and LLaMa3.2-CAD-3B at 0.10.
>
> For both instruction-level and response-level, LLaMa3.2-CAD-1B achieves optimal performance at a higher misleading token threshold than LLaMa3.2-CAD-3B. This suggests that smaller models, which are more sensitive to misleading tokens, benefit from a higher threshold that more effectively filters out disruptive tokens.
>
> **Q2(b)：Were negative values of τ considered?**
>
> **A2(b):** Since the threshold parameter τ represents the percentage of misleading tokens relative to instruction tokens, there is no need to consider negative values.
>
> **Q2(c)：How sensitive is the model's performance to different values of τ across datasets?**
>
> **A2(c):** To evaluate how the choice of τ affects performance, we measured each model’s accuracy on three benchmarks (Math, Code, and SQuAD 2.0) at τ = 0.05, 0.10, and 0.15:
> | Task       | Model                | τ=0.05  | τ=0.10  | τ=0.15  |
> |------------|----------------------|:-------:|:-------:|:-------:|
> | **Math**   | LLaMa‑1B‑Instruct    | 32.43%  | **34.40%**  | 33.17%  |
> |            | LLaMa‑3B‑Instruct    | **51.88%**  | 51.07%  | 50.55%  |
> | **Code**   | LLaMa‑1B‑Instruct    | 17.85%  | 18.27%  | **19.76%**  |
> |            | LLaMa‑3B‑Instruct    | 32.83%  | **33.55%**  | 32.95%  |
> | **SQuAD 2.0** | LLaMa‑1B‑Instruct  | 72.12% | 73.33%  | **74.78%**  |
> |             | LLaMa‑3B‑Instruct   | 88.67%  | **89.44%**  | 83.66%  |
>
> We observe that LLaMa‑1B‑Instruct generally achieves its best performance at \( τ = 0.15\), while LLaMa‑3B‑Instruct consistently performs best at \( τ = 0.10\) across these tasks. This indicates that the threshold \( τ_mis \) is relatively stable in different datasets.
>
> **Q3: Sections 4.1 and 4.2 present experimental results, but it is unclear which model(s) these results are based on. Could the authors specify the model architecture and size used in each subsection to improve reproducibility?**
>
> **A3:** Thank you for the suggestion. The base model used in both Section 4.1 and Section 4.2 is **Llama‑3.2‑3B‑Instruct**, and we will clarify these settings more explicitly in the future version.
>
> **Q4：In Table 1, the proposed methods fall behind some other methods in some datasets—for example, the Instruct Model or KD w/o Mask occasionally outperforms the masking-based methods in LeetCode and Livecode-Bench. Could the authors analyze or explain the possible reasons behind these anomalies?**
>
> **A4:** We attribute these anomalies primarily to **the limited coding capability of LLaMa3.2‑1B‑Instruct**.
> To validate this, we conducted a preliminary experiment using Kodcode’s 5K synthetic LeetCode dataset.  After removing misleading tokens, LLaMa3.2‑1B‑Instruct solved only **264 additional problems (+5.28%)**, whereas **Evol** and **OSS** achieved gains of  **909 (+15.2%)** and **992 (+16.5%)**, respectively—a performance gap of around **10%**.  This clearly suggests that most of the failures by LLaMa3.2‑1B‑Instruct on LeetCode and Livecode-Bench are **due to inherent limitations in its coding and reasoning capabilities**, rather than misleading patterns.  As a result, even improved focus on key information cannot significantly boost its performance.
>
> In contrast, larger models such as LLaMA‑3.2‑3B‑Instruct and Qwen2.5‑Math‑1.5B exhibit **stronger generalization and reasoning capabilities**, which enables them to more effectively learn to disregard misleading patterns and focus on truly informative tokens. This improved attention behavior can then be **successfully transferred to more complex tasks like LeetCode and Livecode-Bench**, leading to more consistent performance gains.

---

> ### Author Response · Authors · 2025-08-05
> **Seeking Further Discussion on Rebuttal**
>
> Thank you again for the great efforts and valuable comments. We have carefully addressed the main concerns in detail. We hope you might find the response satisfactory. As the discussion phase is about to close, we are very much looking forward to hearing from you about any further feedback. We will be very happy to clarify any further concerns (if any).

---

> ### Comment · Area_Chair_co2r · 2025-08-06
> **Official Comment from AC**
>
> Dear Reviewer,
>
> As the Area Chair handling this submission, I would like to express my appreciation for the thorough and insightful comments you have provided. The authors have now submitted a comprehensive rebuttal addressing each of the points raised in your reviews.
> To facilitate a fair and rigorous decision-making process, I kindly urge each reviewer to carefully assess whether the authors' rebuttal adequately addresses your specific concerns and technical questions. Your active participation and constructive input during the upcoming discussion phase will be crucial in arriving at a well-informed decision.
>
> Please feel free to share any additional thoughts or clarifications you may have as we proceed with our deliberations.
>
> Best regards,
>
> Your AC

---

> ### Author Response · Authors · 2025-08-06
> **Looking for Further Discussion and Feedback**
>
> Dear Reviewer 2JsC,
>
> We have diligently addressed every concern raised, especially our **spurious-confounder analysis on larger-scale models** and our **detailed threshold analyses on the Math, Code, and SQuAD 2.0 benchmarks**. As we have been awaiting your feedback **for nearly six days**, we kindly request your review of the rebuttal, we would greatly appreciate any further feedback, suggestions, or concerns at your earliest convenience. We are eager to engage in further discussion and clarify any unresolved points to ensure all matters are fully addressed.
>
> Best regards,
>
> Authors of Submission 9306

---

> ### Author Response · Authors · 2025-08-09
> **Looking for Further Discussion and Feedback– Submission 9306**
>
> Dear Reviewer 2JsC,
>
> We sincerely thank you for your earlier comments and the opportunity to address them.
> During the discussion phase, we have carefully and thoroughly responded to all points raised, particularly our **spurious-confounder analysis on larger-scale models** and **the detailed threshold analyses for the Math, Code, and SQuAD 2.0 benchmarks**.
>
> As the discussion deadline is now less than **12 hours** away, we would greatly value your feedback on our responses. If you feel that your concerns have been addressed, we would be grateful if this could be reflected in your updated evaluation. If there are still outstanding questions, we are more than happy to provide further clarification within the remaining time to ensure a constructive exchange before the rebuttal period concludes.
>
> Best regards,
>
> Authors of Submission 9306

---

### Author Response · Authors · 2025-08-09
**General Response**

We sincerely thank all reviewers for their thoughtful and constructive feedback. We are encouraged by the recognition of our paper’s key contributions and strengths.

In particular, we appreciate the recognition of the **strong motivation and novelty** of our Causal Attention Distillation (CAD) framework, grounded in causal principles and offering a new solution to mitigating LLMs’ susceptibility to misleading tokens (**qdQ9**, **WSL6**). Reviewers also acknowledged the **methodological soundness** of CAD, including the integration of causal inference with gradient analysis to separate false correlations (**2JsC**, **ZMF9**, **WSL6**). The **comprehensive experiments** across multiple settings and benchmarks were positively noted for consistently improving over standard KD (**2JsC**, **qdQ9**, **ZMF9**, **WSL6**), while the **systematic comparison of masking strategies and response segmentations** was recognized for strengthening the empirical claims and justifying the design choices (**ZMF9**). We are encouraged by the recognition of CAD’s **interpretability**—demonstrated through causal graphs, attention heat maps, and a case study illustrating attention shifts toward causally relevant information (**2JsC**, **ZMF9**)—as well as the feedback on the **clarity and organization of our writing**, ensuring accessibility to a broad audience (**2JsC**, **qdQ9**, **ZMF9**).

We have carefully addressed each comment from the reviewers and incorporated targeted updates in our rebuttal. These include **further validation of CAD in the QA domain**, **extensive robustness evaluations**, **detailed threshold analyses** on the Math, Code, and SQuAD 2.0 benchmarks, **comprehensive training data composition statistics**, and **efficiency comparisons with baselines**. We believe these additions not only strengthen our contributions but also directly resolve the concerns raised. In the revised manuscript, we will incorporate these new experiments, analyses, and discussions, and below we summarize the core contributions of our study alongside these updates.
_______

**Core Contributions of Our Work**

* **New Finding** – Across LLaMA and Qwen families, simply removing misleading tokens yields *large gains*—over **20%** in math reasoning and **10%** in code generation—revealing that attention to causally relevant details is critical.

* **Novel Framework** –  a causally grounded, two-stage distillation method that detects misleading tokens via gradient-based teacher–student analysis and teaches the student to ignore them through counterfactual supervision.

* **Strong Results & Interpretability** – Consistent improvements over **CoT KD**: **+2.41%** on math reasoning (GSM8K, MATH, OlympiadBench), **+2.48%** on code generation (HumanEval+, LeetCode, LivecodeBench), and **+4.00%** on QA (SQuAD 2.0), with attention visualizations showing focus shifts toward causally relevant tokens.
-----------
**Updates of experimental results during Rebuttal**

* **Section 3.2 (Main Experiments):** Plan to add *validation of CAD’s effectiveness in the QA domain* as part of the primary experimental results (**Reviewer qdQ9**).
* **Section 4.4 (Main Experiments):** Plan to add *detailed threshold analyses* on the Math, Code, and SQuAD 2.0 benchmarks to justify threshold selection (**Reviewers 2JsC, ZMF9, WSL6**).
* **Section 4.5 (Ablation Study):** Plan to add *results from an ongoing ablation study* that removes original samples to evaluate the importance of retaining all original samples during training (**Reviewer WSL6**).
* **Appendix C.1 (Preliminary Experiments):** Plan to add *spurious-confounder analysis* on larger-scale models to validate the generality of the observed phenomenon (**Reviewer 2JsC**).
* **Appendix D.2 (Robustness Evaluations):** Plan to include *extensive robustness evaluations* of CAD across varying noise levels and masking ratios (**Reviewer qdQ9**).
* **Appendix D.3 (Efficiency Analysis):** Plan to include *detailed time-efficiency comparisons* of CAD versus baseline approaches (**Reviewers ZMF9, qdQ9**).
* **Appendix D.4 (Data Statistics):** Plan to add *comprehensive training data composition statistics* to clarify dataset composition and intention (**Reviewer WSL6**).

**Updates of in-depth discussions during Rebuttal**

* **Section 2.1 (Causal Framing):** Plan to *better align the causal framing with complex token-level dependencies*, clarifying how CAD addresses misleading patterns in both instructions and reasoning steps (**Reviewers ZMF9, WSL6**).

-----------
We believe these additions and clarifications comprehensively address the reviewers’ concerns and further enhance the quality and clarity of our work. We look forward to the reviewers’ favorable consideration and remain grateful for their valuable feedback.

Best regards,

**Authors of Submission 9306**

---

### Decision · Program_Chairs · 2025-09-17

**Decision:**

Accept (poster)

**Comment:**

This paper introduces Causal Attention Distillation (CAD), a two-stage framework that enhances LLMs’ reasoning by identifying and masking spurious tokens through gradient-guided teacher-student comparisons, grounded in causal theory to reduce attention bias toward misleading markers. The man contributions of this work are: 1) it addresses a critical issue of spurious correlations in LLM reasoning, 2) it combines causal principles with practical distillation techniques, 3) it shows consistent performance gains across models/tasks, and 4) it enhances interpretability via clear visualizations. The authors are advised to effectively integrate the constructive suggestions from reviewers into the final version.